# Current water contact and *Schistosoma mansoni* infection have distinct determinants: a data-driven population-based study in rural Uganda

Fabian Reitzug [1], Narcis B. Kabatereine[2], Anatol M. Byaruhanga[2], Fred Besigye[2], Betty Nabatte[2] & Goylette F. Chami [1]✉

Water contact is a key element of the system of human-environment interactions that determine individual exposure to schistosome parasites and, in turn, community transmission. Yet, there is a limited understanding of the complexity of water contact. We characterised patterns and determinants of water contact within the large-scale SchistoTrack study on 2867 individuals aged 5-90 years in Eastern and Western Uganda, employing Bayesian variable selection and advanced statistical modelling. We found a 15-year gap between the population-level peak in water contact (age 30) and infection (age 15) with practically no correlation ($\rho = 0.03$) between individual-level water contact and infection. Adults had higher water contact than children, and 80% of individuals with water contact lived within 0.43 km of water bodies. Domestic water contact was most common for children and women, while occupational water contact was most common for men. Water contact was positively associated with older age, fishing or fish mongering occupations, the number of water sites, and type (beach/pond/swamp), and lower village-level infection prevalence. Only older age and fishing were positively, though inconsistently, associated with infection status/intensity. By providing profiles of at-risk groups, and suitable water contact metrics, our research opens avenues for spatially-targeted interventions and exposure monitoring in endemic countries.

Environmentally mediated pathogens primarily affect populations in tropical low and middle-income countries (LMICs) and are estimated to cause an annual loss of 129,488 million disability-adjusted life years (DALYs) or ~40% of the global infectious disease burden[1]. Within LMICs, the disease burden is concentrated in rural poor communities. Interactions of pathogen exposure and infection acquisition with socio-demographic and environmental factors have made it difficult to isolate determinants of exposure. Here, we focus on schistosomes,

parasitic flatworms that infect over 200 million people globally[2]. Schistosome transmission is complex and driven by human behaviour, accessibility of safe water and sanitation, occupation, and ecological conditions for freshwater snails that are the intermediate host of the parasite. Exposure to schistosomes occurs during water contact with lakes, rivers, or streams through activities including swimming, bathing, or fetching drinking water[3]. During water contact, cercariae—the free-living stage of the parasite—enter a human host by burrowing

[1]Big Data Institute, Nuffield Department of Population Health, University of Oxford, Oxford, United Kingdom. [2]Division of Vector-Borne and Neglected Tropical Diseases, Uganda Ministry of Health, Kampala, Uganda. ✉e-mail: goylette.chami@ndph.ox.ac.uk

through the skin. With no available vaccine, mass drug administration (MDA) using praziquantel has been adopted as the main control strategy by the World Health Organisation (WHO)[4]. However, treatment does not prevent reinfection, and past MDA campaigns have experienced low treatment coverage and missed marginalised households. There are concerns that repeated MDA could lead to drug resistance[5,6]. To achieve the targets set out in the 2030 WHO roadmap for neglected tropical diseases[7], there is a need to complement MDA with additional control interventions such as water access, sanitation and hygiene (WASH) provision, environmental control/modification or behaviour change[8–12]. Yet, the knowledge required for the identification of high-exposure groups and for targeting key factors that determine exposure currently is limited.

Water contact has become a well-established proxy indicator for exposure due to the difficulties in directly measuring cercarial exposure. Previous studies have largely been cross-sectional and used self-reported water contact activities or constructed crude binary indicators of water contact for the purposes of predicting current infection[3]. Among 101 studies in a recent systematic review and meta-analysis by Reitzug et al., only 21.8% (22/101) of studies collected snail abundance data to account for environmental factors relevant for translating water contact into parasite acquisition risk[3]. Attempts to integrate water contact with environmental variables have not consistently improved the ability to predict infection[13–16]. This may be due to limitations in data such as rough estimates of bodily immersion data[13,15], assumptions made about cercarial density[13,16], or methodological limitations, including no systematic variable selection and no out-of-sample validation[3]. Due to regular MDA, studying the role of exposure for infection is further complicated by immunity which can be acquired through past infection or successful treatment with praziquantel[17–19]. Understanding exposure also has proven difficult as no standardised exposure measurement tools exist, and there is high heterogeneity across existing studies[3].

There is a lack of comparative studies on predictors of exposure versus current infection. It has been found that infection prevalence varies based on household distance to waterbodies[20,21] but whether this is explained by corresponding trends in exposure over distance remains unclear. Applied statistical models of exposure have predominantly focused on water contact as a predictor of infection without characterising water contact as its own outcome[3]. For estimating the force of infection, current mathematical models infer age-specific water contact levels from infection trends over age[22,23]. These models typically assume that age-specific infection levels are a non-linear function of cumulative water contact up to that age and the survival of adult schistosomes within the host. There is a need to characterise key determinants of water contact and assess whether infection and water contact follow similar trends over age.

We comprehensively characterised water contact using applied statistical models. Data was collected from January to February 2022 on the River Nile, Lake Albert, and Lake Victoria in rural Uganda as part of a population-based study within the baseline of the SchistoTrack cohort[24]. Across 38 diverse villages in Pakwach, Buliisa and Mayuge Districts, we surveyed 2867 individuals aged 5–90 from 1444 randomly sampled households. Socio-demographics, biomedical information, WASH information, and environmental data were collected. To measure exposure, individuals were asked whether they engaged in any of 11 different water contact activities (fishing, swimming, getting drinking water, etc.) and their respective weekly frequency and duration. We defined any water contact as engaging in at least one water contact activity per week and grouped 11 activities into broader categories (occupational, recreational, and domestic water contact). *Schistosoma mansoni* infection status was ascertained using Kato-Katz stool microscopy[25] and point-of-care circulating antigen tests[26]. Malacological data on snails, as well as waypoints of water sites, households, schools, and village centres, were used to capture relevant environmental and spatial variables. We answered the following questions. What are the major human-environmental determinants of water contact, and at which level (individual, household, or village) is the influence of those determinants? What are (if any) the shared dimensions of water contact determinants with infection determinants?

## Results

### Water contact type, timing, frequency, and duration

A study overview is presented in Fig. 1. Detailed variable definitions and characteristics of the 2867 participants are presented in Supplementary Tables S1 and 2. Among study participants, 46.7% (1339/2867, 95% confidence interval (CI) 44.9–48.5%) reported that they had water contact with open freshwater bodies at least once per week. The median frequency of water contact was six times per week (interquartile range (IQR) 3-11), and the median duration in hours per week was eight (IQR 3.5–17.5). Both frequency and duration exhibited overdispersion, with only a few individuals engaging in high-duration or high-frequency water contact (Supplementary Fig. S1). The three most common water contact activities among participants were getting drinking water (17.3%, 497/2867), washing clothes with soap (16.8%, 481/2867), and fishing (12.3%, 354/2867, Supplementary Table S3). Washing clothes with soap was as likely as washing clothes without soap (5.8%, 28/481, versus 5.2%, 124/2386, $p = 0.67$). Overall, water contact was most commonly conducted between 6–9 am, with 23.6% (956/4046) of contacts taking place at this time. The typical time of day of water contact varied by activity type. Washing clothes with soap and getting drinking water were most commonly done after sunrise (6–9 am), with 32.8% (163/497) and 34.1% (164/481) of water contacts related to these activities, respectively, occurring during this time. For fishing, the early evening (5–7 pm) was the most common time accounting for 24.3% (86/354) of fishing water contacts. Concerning frequency of water contact, the median number of weekly trips was two for washing clothes with soap (IQR 2–5), seven for getting drinking water (5–14), and four for fishing (IQR 2–7). Among these activities, the median duration of water contact was highest for fishing at four hours (IQR 3-4). Washing clothes with soap or drinking water had a median duration of only one hour, noting 30 min as the minimum amount of time recorded in the study survey.

### Water contact dependence on distance to water sites

Households were located within 0.01–2.17 km (mean 0.35 km, 95% confidence interval (CI) 0.34–0.36 km) of the River Nile, Lake Albert or Lake Victoria (Fig. 2b–d). Household distance varied by district. In Pakwach District, the mean distance to the River Nile or Lake Albert was 0.41 km (95% CI 0.39–0.43 km); whereas in Buliisa District, the mean distance to Lake Albert was 0.24 km (95% CI 0.23–0.26 km). Household waterbody distances were more similar between Mayuge and Pakwach although they were the districts furthest apart. Water contact varied by waterbody distance; in households residing directly on the shoreline (≤100 m), 53% of individuals (315/589) had water contact. In households located >1 km from a water site, 26% (36/137) of individuals reported having water contact. Water contact declined rapidly with shoreline distance as 80% per cent of all individuals with water contact lived within 0.43 km Euclidean distance of water sites (Supplementary Fig S2). Among individuals within 0.43 km, 50.3% had water contact, whereas only 36.3% of individuals living further than 0.43 km away had water contact. Using generalised additive models (GAMs), we modelled the decline in water contact over household distance to the closest water site. For every 100 m-increase in the distance between 0-1 km, we found a 3.4%-point reduction in the proportion of participants with water contact. Consistently decreasing gradients were observed across domestic, occupational, and recreational activities (Fig. 3a). Gradients in water contact differed depending

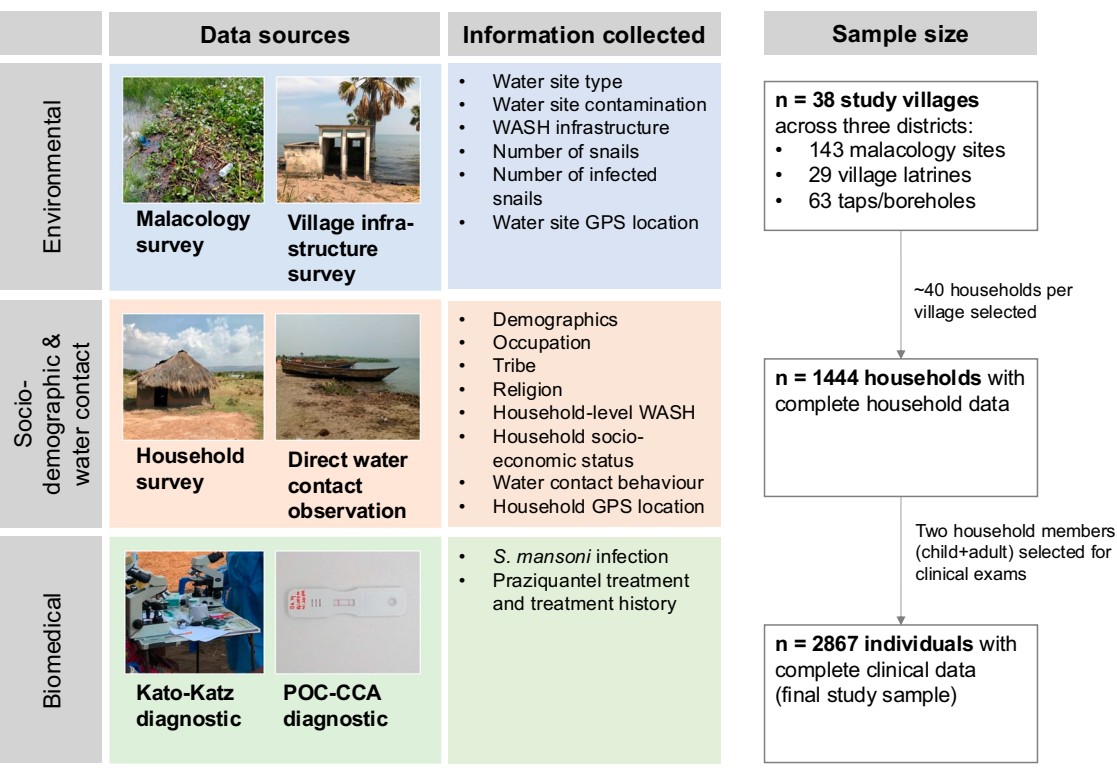

**Fig. 1 | Study overview.** Data sources, information collected, and sample size of the SchistoTrack study. Abbreviations: WASH = water, sanitation, and hygiene. GPS = global positioning system.

on the distance metric used (Fig. 3b–d). Village centre distance to the closest water site produced weaker gradients than household distance with a 1.9% versus 3.4%-point reduction for every 100 m-increase with village distance compared to household distance (Fig. 3c). When using primary school distance to the closest freshwater body, there was no relationship between distance and the percentage of households who engaged in water contact (Fig. 3d).

**Gender and age-specific water contact patterns**
We used GAMs to examine gender and age-specific patterns of water contact. Overall, 49.3% (775/1573) of females and 43.6% (564/1294) of males reported water contact. The frequency of water contact was higher for females than males at a median of seven versus five trips per week (Wilcoxon rank sum test $p < 0.01$, Supplementary Table S4). Conversely, females had a lower duration of water contact than males, with seven compared to ten hours reported per week (Wilcoxon rank sum test $p < 0.01$, Supplementary Table S5). Relationships between current water contact with gender and age were nonlinear and varied substantially over the life course. The modelled age-dependent proportion of participants with water contact was 16% (95% CI 14–18%) at age five, peaked at 70% (95% CI 66–74%) at age 30, and declined to 28% (95% CI 20–36%) at age 70 (see Fig. 4a). Across ages 18–35, the proportion of individuals with water contact was higher for males than females (Supplementary Fig. S3). Beyond any water contact, the type of water contact activity was gender-dependent (Supplementary Table S3). For instance, 24.3% (315/1294) of males reported going fishing, while the same figure was only 2.5% (39/1573) among females ($\chi^2 = 311.56$, $p < 0.01$). Collecting drinking water was reported by 22.1% (348/1573) of females but only 11.5% (149/1294) of males ($\chi^2 = 55.02$, $p < 0.01$). Washing clothes with soap was also more common among females than males (23.4% (368/1573) versus 8.7% (113/1294), respectively, $\chi^2 = 108.27$, $p < 0.01$). The proportion of water contacts taking place during peak cercarial shedding time, defined as between 10 am–3 pm[27], was not significantly different between males and females

for any activity. However, predominantly female activities−getting drinking water, washing clothes with soap, and washing jerry cans or household items−were more likely to be conducted during peak shedding time than male activities such as fishing (Supplementary Table S6). Gender differences in water contact frequency, duration and dominant activity type became pronounced in adolescents between ages 15−19 (Fig. 4b, c). Until age 15, water contact patterns of both females and males included mostly domestic activities. After age 15, females maintained high levels of domestic water contact while the relative involvement of males in domestic activities strongly declined. Among adults (age 18 +), domestic water contact accounted for 75.3% of total water contact duration for females, while occupational water contact accounted for 81.8% of total water contact duration for males (Fig. 5).

**Relevant human-environmental dimensions of water contact**
We used Bayesian variable selection (BVS) on a comprehensive set of 27 socio-demographic, biomedical, water, sanitation, and hygiene (WASH), and environmental candidate variables (Fig. 6a–d). Variables with inclusion probabilities ≥0.5 were used in multivariable logistic regression models to predict water contact. Selected socio-demographic factors were age, age², gender, and occupation. One village-level WASH variable −the distance to the closest public latrine from the household− was included, as well as village infection prevalence. Four environmental variables were selected, including the type of water site closest to the household, the closest water site within the village, the number of water sites per village, and faecal site contamination occurring at the closest site to the household. To summarise which variable types and levels were relevant for predicting water contact, we calculated the proportion of variables by group and level, as weighted by their respective inclusion probabilities (Fig. 6e, f). Among selected variables, socio-demographics accounted for 47% of all weights, followed by environmental variables at 39%, WASH at 7%, and biomedical variables at 7% (Fig. 6e). Individual-level factors

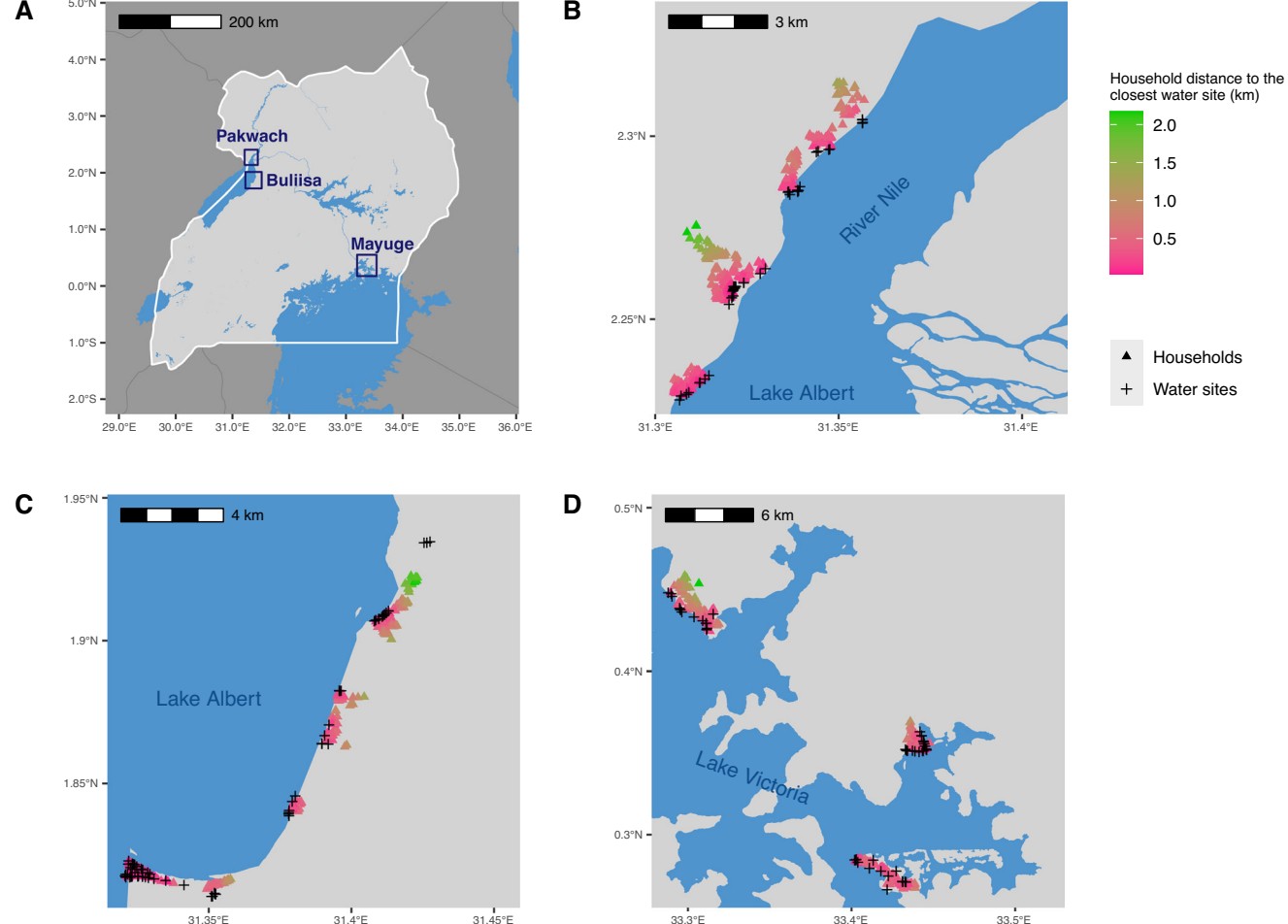

**Fig. 2 | Study areas.** The locations of 2867 study households (randomly displaced by 50 m) and 143 water sites sampled by malacologists are shown. Waterbody boundaries were taken from the open resource dataset from the World Resources Institute. **A** Locations of the three study districts Pakwach, Buliisa, and Mayuge in Uganda. **B** Zoomed in view of Pakwach. Households and water sites are shown along Lake Albert and the River Nile. **C** Zoomed in view of Buliisa. Households and water sites are shown along Lake Albert. **D** Zoomed in view of Mayuge. Households and water sites are shown along Lake Victoria. Household distance in this figure represents the Euclidean distance of the household location to the closest water site.

accounted for 30%, household-level factors for 55%, and village-level factors for 15% of all weights (Fig. 6f).

**Determinants of any water contact**

Figure 7 presents the main model of any water contact. Significant predictors of current water contact spanned socio-demographic, bio-medical, WASH, and environmental variables. Both age (OR 1.18; 95% CI 1.15–1.21) and age$^2$ (OR 0.9978; 95% CI 0.9975–0.9982) were sig-nificantly associated with water contact. The relevance of age$^2$ sup-ported a declining relevance of older age with water contact, consistent with the exploratory GAMs. Females had 1.41 times higher odds of having water contact compared to males (95% CI 1.17–1.68). Fishing and fishmongering occupations were associated with 6.67 times (95% CI 4.06–10.95) and 2.10 times (95% CI 1.12–3.95) higher odds of water contact, respectively when compared to other or no income-earning occupations. Occupation and gender were sig-nificantly associated ($\chi^2 = 400.95$, $p < 0.01$). When occupation was removed from the model, the gender coefficient became insignificant (OR 1.05; 95% CI 0.89–1.24, Supplementary Fig S4). Observed faecal contamination at the water site closest to the household was asso-ciated with 24% lower odds of having water contact (OR 0.76, 95% CI 0.63–0.92) when compared to water sites with no such contamination. Each 1 km increase in distance to the closest public latrine was asso-ciated with 13% lower odds of having water contact (OR 0.87; 95% CI

0.81–0.94). Yet, latrine distance was significantly correlated with dis-tance to the closest water site ($\rho = 0.33$, $p < 0.01$). Water site ecology influenced water contact behaviour. Individuals who lived closest to a beach, swamp, or pond had significantly higher odds of having water contact compared to people living closest to a river (ORs 1.57–3.24). Access to water sites, measured by the number of water sites per vil-lage (OR 1.13; 95% CI 1.05–1.21) and whether the water site closest to the household was within the village (OR 1.49; 95% CI 1.25–1.78), was positively associated with the odds of water contact. In villages with ≥50% *S. mansoni* prevalence, individuals had 30% lower odds of engaging in water contact compared to villages with 10–49% pre-valence (OR 0.70; 95% CI 0.57–0.87). After adjustment for covariates, there were no significant regional differences in water contact when comparing the Western districts of Pakwach and Buliisa to Mayuge.

Separate logistic regression models predicting domestic, occu-pational, and recreational water contact showed that the effect of gender varied between activity types (Supplementary Fig. S5). Females had 2.43 times higher odds of engaging in domestic water contact than males (OR 2.43; 95% CI 1.97–3.00), but 72% and 83% lower odds of having recreational and occupational water contact (OR 0.28; 95% CI 0.11–0.73 and OR 0.17; 95% CI 0.11–0.27, respectively). Negative bino-mial models predicting water contact frequency and duration showed no significant gender differences in duration and frequency after adjusting for covariates (Figs. S6, 7).

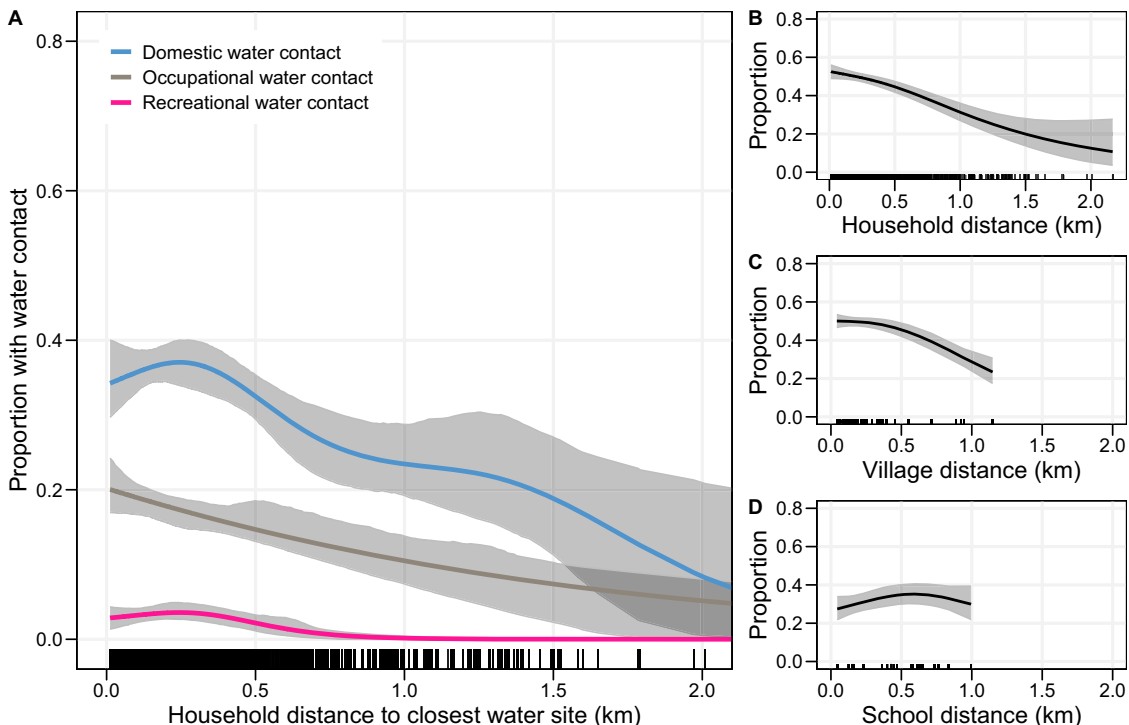

**Fig. 3 | Proportion of individuals with water contact over household waterbody distance.** The proportion of individuals with current water contact over Euclidean household distance to the closest water site (in km) was modelled using generalised additive models. **A** Proportion of individuals with domestic, occupational, and recreational water contact. **B**–**D** Comparison of lake gradients in water contact across alternative distance metrics: household distance, village centre distance, and school distance to the closest water site. **B** Proportion of individuals with any water contact by household-level distance to the closest water site. **C** Proportion of individuals with water contact by village centre distance to the closest water site.

The village centre location was identified as the cultural centre by the village chairman. All households within a village were assigned the distance from the village centre to the closest water site. Compared to household water site distance with a maximum of 2.2 km, the village centre distance distribution was truncated at 1.2 km as the furthest village centre in this study was 1.2 km away from the closest water site. **D** Prevalence of water contact by distance from the school closest to the household to the nearest water site. The sample was restricted to $n = 597$ enrolled children. The 95% confidence intervals in Panel (A) were computed via bootstrap with 1000 repeats.

## Determinants of water contact versus infection

We compared determinants of water contact and schistosome infection status, defined as ≥1 egg per gram (EPG) of stool. Overall, 43.3% (1240/2867) of study participants were infected with *S. mansoni*, and 8.2% (236/2867) were heavily infected (400 + EPG). District-level prevalence was highest in Pakwach (50.4%, 477/947), followed by Buliisa (44.1%, 422/958), and Mayuge (35.4%, 341/962). Modelled *S. mansoni* infection prevalence varied over age; 29% (95% CI 25–33%) of children were infected at age five, prevalence rose to a peak of 63% (95% CI 59–67%) at age 15 and then declined to 31% (95% CI 24–36%) at age 50 (Fig. 4a). This trend differed from age-dependent water contact which peaked much later at age 30 and remained at a significantly higher level than infection until age 65. Modelled age-specific infection prevalence trends in GAMs were similar between participants with and without water contact (Supplementary Fig. S8). Infection prevalence did not follow the same linear decline over household distance from water sites which was observed for water contact (Supplementary Fig. S9). Even crude correlations between current water contact and infection status showed no association ($\rho = 0.03$, $p = 0.11$, obs. 2867). Pairwise correlations of infection status ($\rho = 0.16$, $p < 0.01$) or water contact ($\rho = 0.08$, $p = 0.01$) between adults and children within the same household were weak (obs. 2867). Village-level infection prevalence was uncorrelated with the village-level proportion of individuals with water contact ($\rho = 0.004$, $p = 0.98$, obs. 38).

We selected predictors of infection status using the same methods and candidate set as for water contact (plus six additional water contact variables for the infection model, Supplementary Table S1). The number of variables selected for water contact was ten compared to seven for infection status. Among these 10 variables for water contact, only 50% were selected for inclusion in the infection status model. Shared variables were age, age², occupation, number of water sites per village, and village infection prevalence. Gender was not selected for the infection model. No water contact or WASH variable met the threshold of inclusion for infection status. Other predictors of water contact which were not selected for infection included faecal contamination of the water site closest to the household, type of water site closest to the household, closest water site in the village, and distance to the closest public latrine. There were only two predictors of infection which were not selected for the water contact model: education level attained and the landform type of the water site in the village. When grouped by variable type, socio-demographics (accounting for 62% of variables) and environmental variables (25%) were most relevant for infection (Fig. 6e). Socio-demographics were more relevant for infection status than for water contact, while environmental variables were less important for infection status compared to water contact. When compared by level, household-level variables were comparatively less relevant for infection status than for water contact, while individual-level variables were more relevant for infection status than for water contact (Fig. 6f).

Figure 8 shows the results of logistic regressions predicting infection status and heavy infection. Among the five variables consistently selected for water contact and infection, no variable except for age² was significant and of similar magnitude, defined as having overlapping CIs, between the models. Age was positively associated with both outcomes, but the effect size of each one-year increase in age was of greater magnitude for water contact (OR 1.17; 95% CI 1.15 – 1.21)

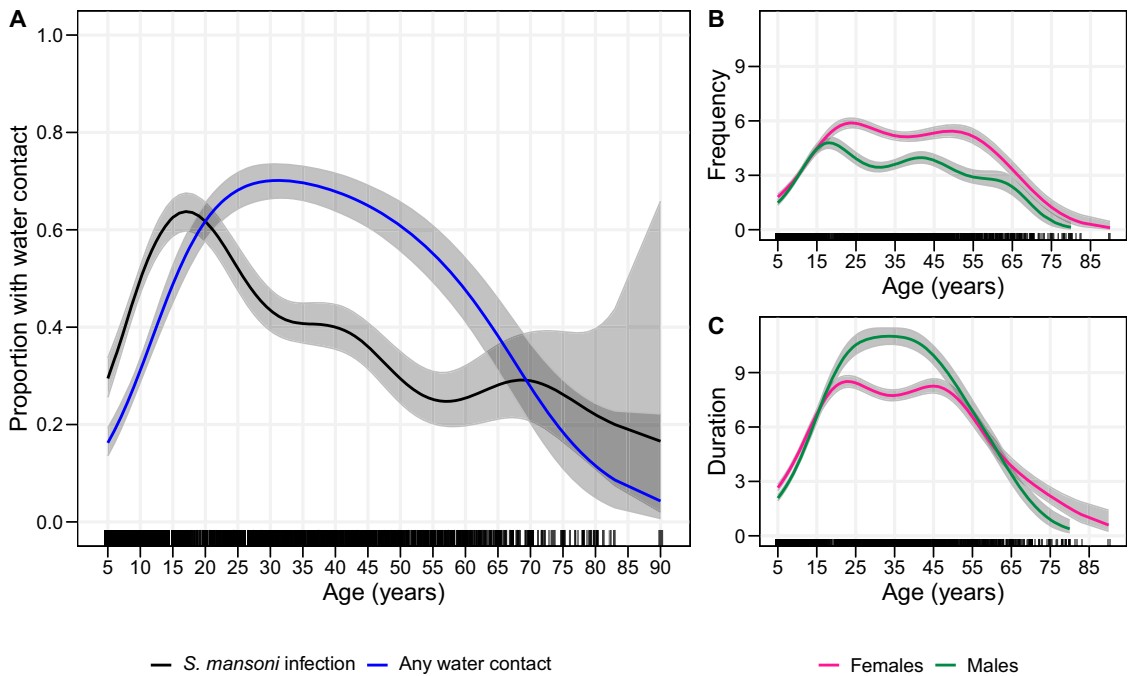

**Fig. 4 | Variation in water contact and infection over age. A** Proportion of individuals with current water contact and *S. mansoni* infection prevalence over age modelled using generalised additive models. **B** Frequency of water contact measured by the weekly number of trips to waterbodies over age and gender. **C** Duration of water contact measured in numbers of hours per week, over age and gender.

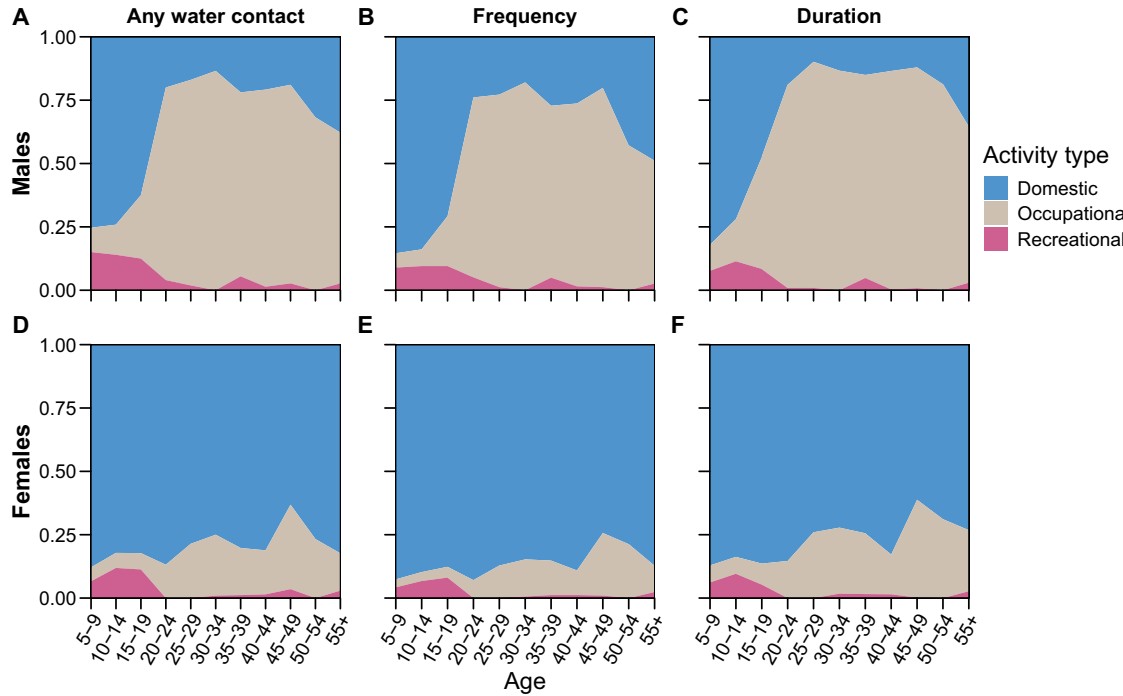

**Fig. 5 | Composition of water contact activities over age and gender.**
**A** Proportion of all water contacts within each 5-year age group which were domestic, occupational, or recreational over age for males. **B** Frequency, measured as the proportion of trips to waterbodies that were domestic, occupational, or domestic over age for males. **C** Duration, measured as the proportion of water contact time which was accounted for by domestic, occupational, or recreational activities over age for males. **D** Proportion of all water contacts within each 5-year age group which were domestic, occupational, or recreational over age for females. **E** Frequency, measured as the proportion of trips to waterbodies that were domestic, occupational, or domestic over age for females. **F** Duration, measured as the proportion of water contact time which was accounted for by domestic, occupational, or recreational activities over age for females.

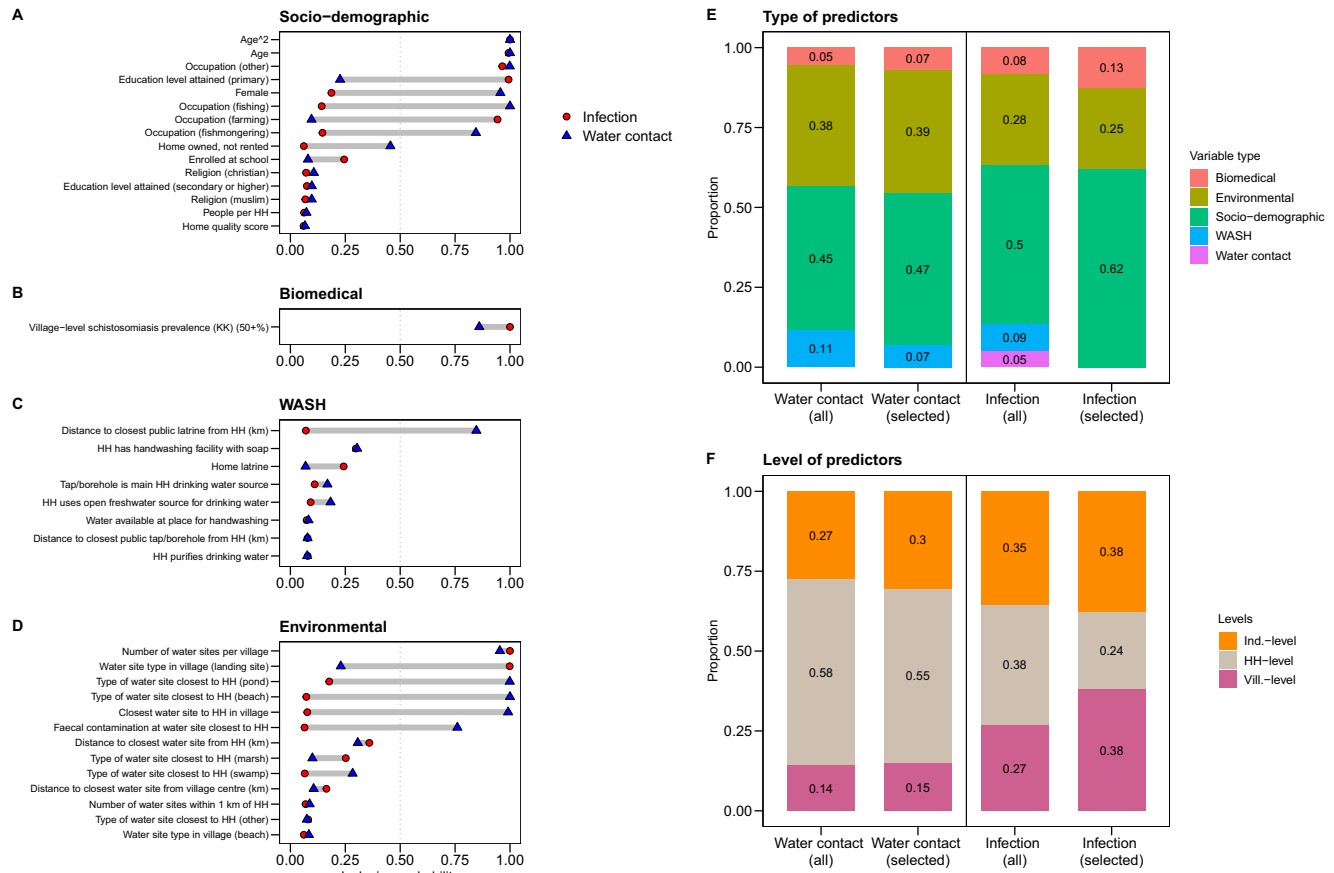

**Fig. 6 | Relevant dimensions of exposure and infection.** Abbreviations: Prop. = proportion. HH = household. WASH = water, sanitation, and hygiene. Ind.-level = individual-level. Vill.-level = village-level. Panels (**A**–**D**) depict the relevance of different types of variables for predicting water contact and infection, as indicated by their inclusion probability in the 'best predictive model' (median probability model) based on Bayesian variable selection (BVS). Socio-demographic variables. **B** Biomedical variables. **C** WASH variables. **D** Environmental variables. The dashed vertical lines in (**A**–**D**) represent the cut-off (pr ≥ 0.5) used to select variables for inclusion in the final regression models. Panels (**E**, **F**) regroup variables in (**A**–**D**) according to their type and level and show the importance of different levels and types of variables as indicated by their proportional representation in the variable set (weighted by their respective inclusion probabilities). **E** Type of predictor variables. **F** Level of predictor variables. Within each box in Panels (**E**, **F**), the left column shows the proportion each variable category accounts for within the candidate variable set, and the right column shows the importance of each category within the set of selected variables.

than infection (OR 1.05; 95% CI 1.03 – 1.08). There was no evidence of association between the two models; we observed no significant correlation between residuals of water contact and infection models ($\rho = 0.01$, $p = 0.44$, obs. 2867). Even within infection models, there was instability across significant variables (Fig. 8). Fishing was inconsistently associated with infection showing only a relationship with heavy infection intensity (400 + EPG) and not infection status (≥1 EPG). Having attained primary education was associated with higher odds of infection but not with higher odds of heavy infection. All results remain robust even when infection intensity is not used as a binary variable and runs continuously in a negative binomial regression (Supplementary Fig. S10).

### Validation of water contact and infection measurements

We investigated water contact misclassification in our survey data by investigating patterns in direct water contact observations. Directly observed water contacts at the most used sites within study villages were available for ~32% (12/38) of study villages and collected at the same time as the self-reported data. For the age structure of water contacts, the self-reported water contact data closely resembled observed community-wide water contact patterns from direct observation (Supplementary Fig. S11). To address the possibility of missed light infections due to the low sensitivity of Kato-Katz microscopy, we recoded negative Kato-Katz results as positive when POC-CCA

diagnostic results were positive where the trace was classified as negative. Infection results remained similar after reclassification of infection status (Supplementary Fig. S12).

### Predictive capacity of water contact models by variable selection technique

We assessed the predictive performance of the water contact and infection models using 10-fold cross-validation and quantified the influence of the variable selection method (BVS versus likelihood ratio tests (LRTs)) and the relevance of including additional snail and water contact variables (Fig. 9). The water contact model had better predictive performance than the infection status model with an area under the receiver operating curve (auROC) of 0.777 versus 0.693 ($p < 0.01$, Fig. 9a). Neither the inclusion of additional snail variables nor the inclusion of more granular water contact variables substantially improved the predictive accuracy of any model (Fig. 9b, c, see "Methods" for details). BVS outperformed LRTs for water contact prediction (auROC 0.777 versus 0.522, respectively, $p < 0.01$, Fig. 9d and Supplementary Fig. 14). Compared to water contact, variable sets selected for infection status were more comparable between BVS and LRTs (Supplementary Table S6), and the gap in performance was smaller (auROC 0.696 and 0.648, $p < 0.01$, Fig. 9d). There was weak residual spatial correlation in the water contact model (*Moran's I* = 0.064, $p < 0.01$).

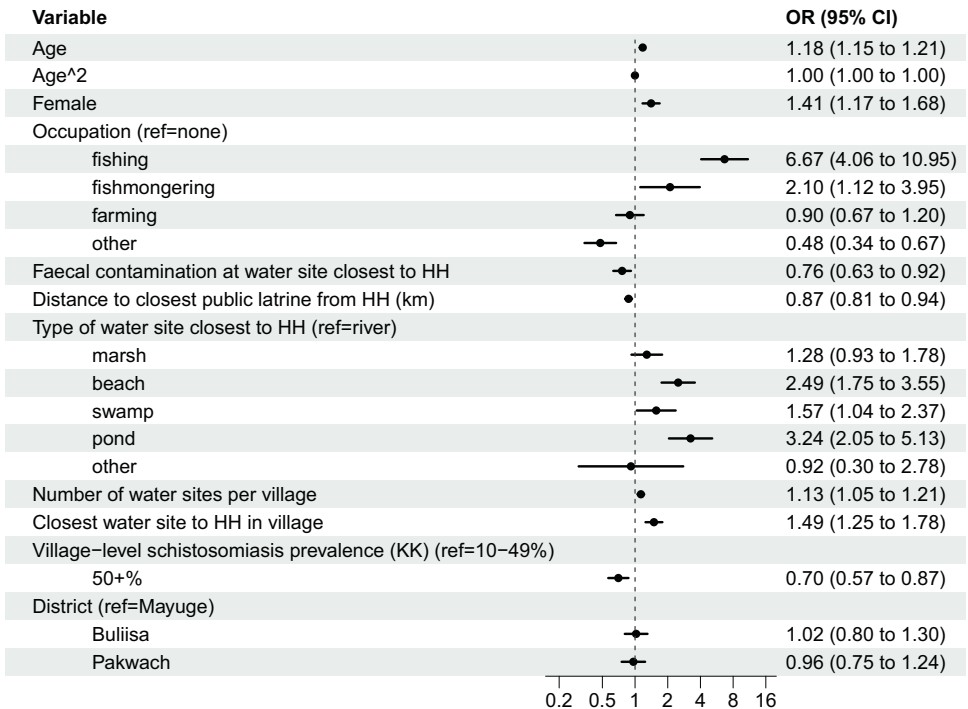

**Fig. 7 | Determinants of water contact.** Abbreviations: Prop. = proportion. HH = household. KK = Kato-Katz stool microscopy. A logistic regression model is shown for current water contact (*n* = 2867) with 95% confidence intervals from standard errors clustered at the household level. The *X*-axis is on a log scale. The global *Moran's I* of the model residuals was 0.064 (95% CI 0.0639–0.0644, *p* < 0.01).

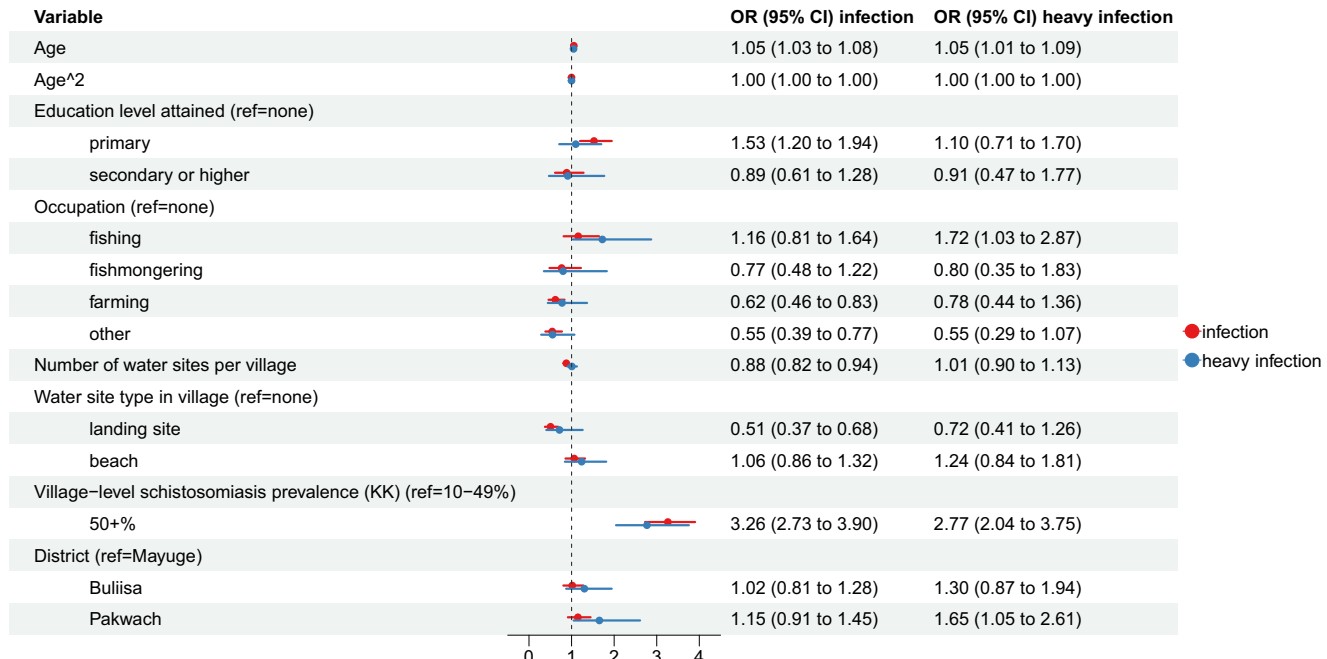

**Fig. 8 | Determinants of infection status and heavy infection intensity.** Abbreviations: KK = Kato-Katz stool microscopy. Separate logistic regression models predicting *S. mansoni* infection and heavy *S. mansoni* infection (*n* = 2867) with 95% confidence intervals from standard errors clustered at the household level. The *X*-axis is on a log scale. The global *Moran's I* of the model residuals was 0.0049 (95% CI 0.0047–0.0052, *p* = 0.39).

## Discussion

The 2022 WHO guideline on schistosomiasis control and elimination, as well as recent cross-NTD guidelines, emphasise the need to complement MDA with behaviour change and WASH to achieve and sustain reductions in schistosome infection prevalence[4,28]. However, this requires an in-depth understanding of the determinants of water contact. Here, a data-driven, population-based analysis was conducted of 2867 individuals within a community-based cohort, SchistoTrack[24],

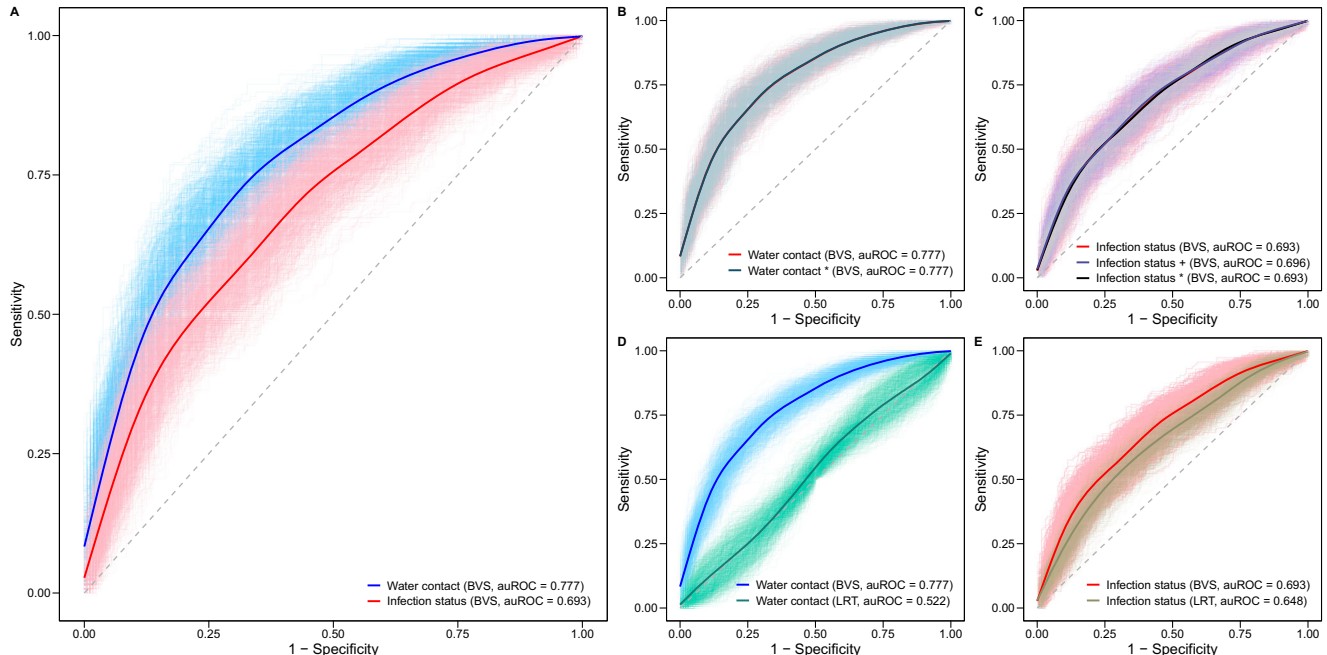

**Fig. 9 | Predictive capacity of water contact and infection models.**
**A** Comparison of auROC for the water contact versus infection status from models in Figs. 7–8 with variables selected using BVS. **B** Comparison of the main water contact model versus a more granular water contact model allowing for the selection of additional variables from a set of seven snail variables, water contact * model, selected using BVS. **C** Comparison of auROC for the infection status model in Fig. 8 versus auROC of two more granular infection status models, the infection + model and the infection * model (Supplementary Figs. S15, 16). In the infection + model variable selection process, we allowed for the selection of additional variables from a set of 82 granular exposure variables using BVS. In the infection *

model variable selection process, we allowed for the selection of additional variables from a set of seven snail variables using BVS. **D** Comparison of auROC for the water contact model selected using BVS versus a water contact model with variables selected using likelihood ratio tests (LRTs). LRTs compared one variable at a time against an 'empty' model with village-level fixed effects at $p < 0.05$ to select variables from the same candidate set as BVS. Differences between variable sets selected via BVS and LRTs are shown in Supplementary Fig. S14. **E** Comparison of auROC for the infection status model using BVS versus an infection model with variables selected using LRTs. All auROCs were computed using 100 repeats of 10-fold cross-validation.

in 38 diverse villages in three districts in Uganda. This study characterised complex water contact and *S. mansoni* infection patterns over age and gender. We developed statistical models and a generalisable analysis pipeline for the wider research community to untangle the intricate dynamics of water contact and infection. Our study included a comprehensive set of exposure variables. We focused on current water contact using detailed individual, household, and environmental predictors of water contact as well as WASH variables. We found that current water contact was not strongly associated with current schistosome infection, highlighting the complex relationship between exposure and infection. Our findings should not be misconstrued as implying that water contact is irrelevant for infection, as all infections are caused by contact at some point with infested freshwater sources.

There was no significant village-level correlation between the proportion of individuals with water contact and village-level infection prevalence. Hotspots exist in Uganda and elsewhere and are defined as high prevalence areas (≥50% by Kato-Katz microscopy) that are unresponsive to repeated MDA[29,30]. Our study was conducted in villages with ≥10% infection prevalence in districts that each had received over 13 annual rounds of MDA since the year 2003[31]. Individuals in hotspot villages (≥50% infection prevalence) had lower odds of water contact when compared to villages with 10–50% prevalence. There have been calls for intensified control in hotspot settings, including the use of exposure reduction interventions[32]. Our findings suggest that such intensified measures, if intended to reduce water contact, should be targeted not only in hotspots but also in moderate prevalence settings.

Age-specific infection patterns did not approximate age-specific water contact suggesting an important role of acquired immunity in explaining infection patterns over age. Despite our population being

from three districts with varying local climates, geography, and tribal and religious groups, water contact was comparatively low in children across all districts. Infection prevalence peaked at age 15—more than a decade earlier than water contact, which peaked at age 30. Our finding is consistent with cross-sectional studies conducted before widespread MDA in endemic settings that reported lower levels of reinfection in adults despite higher levels of water contact[33–35]. The discrepant age trends of water contact and infection support the slow build-up of immunity, possibly due to immunity acquired through repeated exposure to antigens from the death of mature flukes, which may reduce susceptibility to reinfection[36,37]. Given low levels of water contact among children compared to adults, we speculate that even longitudinal water contact data would be unlikely to reveal much more substantive correlations between measures of cumulative water contact and infection. Our study suggests that the relationship between water contact as an element of exposure and acquired immunity associated with water contact is likely highly non-linear; the potential critical threshold of exposure required to explain the sudden drop in infection levels at age 15 years would be suspected to be incredibly low. In our study, we were not able to rule out the role of the timing of the first infection, longevity of single infections, influence of co-infections and other immune-modulating factors, or age-specific immune responses[20,38]. There is a need for future longitudinal studies which combine the collection of exposure histories with immunological data. In terms of modelling, this study advocates for relaxing assumptions about a direct correspondence between current age-specific water contact rates and infection prevalence.

We observed clear gender differences in the dominant types of water contact among adults. Female adults mostly had domestic water contact, while male adults primarily engaged in occupational water

contact. These activities typically take place at different times of the day and involve varying levels of bodily immersion, which may lead to differing exposure levels. A systematic review suggested that gender differences in infection prevalence might be explained by these varied exposure patterns[39]. While the review was unable to account for age differences due to a lack of age-disaggregated reporting, our study emphasises the need for age stratification. We found that water contact patterns were similar in males and females up to age 15. Among adults, being female was positively associated with water contact, but only after adjusting for occupation, underscoring the interrelatedness of gender and occupation. When using a composite outcome, such as any water contact, it remains challenging to isolate the effects of gender. However, by separately modelling different water contact activities, we found that females had significantly higher domestic water contact but significantly lower occupational and recreational water contact. To promote gender equity in exposure reduction programmes, both domestic and occupational water contact should be addressed, as they represent the primary sources of exposure for females and males, respectively.

Past research has tried to separate environmental and human-behavioural elements of exposure[40,41], yet these factors are intricately linked. In our study, environmental features influenced both water contact behaviour and environmental hazards, where the hazard is defined as the local abundance of schistosome cercaria. Here, we identified several environmental factors, such as the number of water sites within the village, site type, and faecal contamination, which impacted water contact patterns. Some of these factors, especially faecal contamination, may also impact environmental hazards. When we included several snail-related factors, such as the number of snails or infected snails at the nearest site, we found that they did not influence water contact behaviour as would be expected given that they cannot be easily observed by individuals. Understanding which ecological factors influence water contact, environmental hazard, or both, can shed light on the complex human-environmental processes involved in exposure and clarify how environmental factors should be most appropriately integrated into exposure indices and parameterised in models.

Distance to waterbodies has been used for approximating the force of infection, sampling individuals for infection studies, and choosing where to implement MDA[14,20,42,43]. Many studies have used binary indicators of distance with varying cut-offs ranging from 0.5–5 km to predict infection[20,21,43–45]. We found a clear, almost linear gradient of the distance of a home to freshwater bodies with water contact. The proportion of participants with water contact declined by 34%-points as household distance increased from 0-1 km to the closest water site. Infection prevalence showed a weaker decline of only 19% from 0-1 km for household distance to water sites, which appeared nonlinear with a much less discernible trend than water contact. In our statistical models, waterbody distance was not selected because the inclusion of occupation, water site availability and other distance-related factors were able to account for spatial patterns in water contact and infection, as indicated by low residual spatial autocorrelation. Different distance measures have been used interchangeably elsewhere, such as household distance[21,44,45], or village distance[46]. We showed that these measures were not equivalent, as we found no water contact gradients in school-based measures. Thus, only household or village distance to large freshwater bodies should be used to estimate exposure gradients. And, critically, whether to use the proportion of individuals with current water contact or infection prevalence as outcomes for estimating distance gradients should be carefully considered in future studies and interventions dependent on the use case.

A key challenge for exposure studies has been reliable water contact measurement. Self-reported measures have been the most widely used[3] but are considered less reliable than direct observations

or wearable global positioning system (GPS) logger data[47,48]. Here, we validated the use of self-reported water contact data by comparing it with direct observations, demonstrating that surveys provide a cost-effective way to obtain valid population-level water contact measures. Still, there remains a need for future water contact ascertainment studies that match individuals directly with more granular measurements, and when such ascertainment is completed, we suggest rerunning our models with regression dilution to investigate the influence of possible measurement error. For future research, more granular measurements of frequency and duration of water contact should be investigated using wearable GPS loggers.

Due to the complexity of water contact, we suggest that appropriate measurement of water contact should be tailored to the specific goal of a national programme or research study. For instance, for monitoring and surveillance surveys, especially when conducted alongside MDA, collecting age, gender, and occupation status may be sufficient to ascertain basic risk factors for water contact. Home location data could be collected to delineate a buffer area in which individuals are at high exposure risk. By contrast, sentinel surveillance sites based at schools are unlikely to identify fine-scale gradients in water contact. For interventions, data on common water contact activities and their typical time of day could aid in identifying target groups and behaviours. With finer spatiotemporal measurements that might include indicators from wearable GPS loggers, in the future, it will be possible to identify the exact durations and intensity of water contact, seasonal and diurnal water contact patterns, and explore the role of human mobility for schistosome transmission.

We found that human-environmental interactions were highly group-specific, suggesting key target groups for WASH. Domestic activities were the dominant type of water contact for children and women. Additional research is needed to target domestic water contact activities such as doing laundry and getting drinking water with safe water infrastructure and evaluate the water contact reductions resulting from such interventions. Despite the inclusion of detailed household-level WASH variables, only distance to public latrines was significantly associated with water contact. Surprisingly, the use of safe water sources and household water purification were not relevant for water contact. Given water contact was dependent on occupation for men, there was no clear suggestion that WASH would influence water contact for this group. Future studies might instead focus on socio-economic interventions, such as cash supplements or employment retraining, to understand how to reduce water contact for men. To test standard approaches in this study, we focused on occupational fishing as opposed to recreational/irregular fishing. If a more comprehensive view of any fishing was undertaken, it may be possible for future research to identify interventions that may shift fishing activities to less risky times of day or to modify the predominant type of fishing activity, using protective equipment such as gumboots, or building safe landing sites.

The results of this study provide a compendium of water contact metrics and patterns to facilitate multifaceted control of schistosomiasis. We found that in our Ugandan setting, children were highly susceptible to infection despite relatively low levels of water contact, providing support for continued MDA. Given the important role of adults in community-level water contact, exposure reduction interventions to reduce water contact should be targeted at all age groups, even in cases where MDA is limited to school-age children. Our study supplements the current WHO guideline for schistosomiasis treatment[4], which defines at-risk communities and groups based on infection prevalence and intensity, by profiling priority groups with high water contact. Concerning the guidelines for selecting implementation units for MDA, water contact was heavily concentrated in households within 0.43 km of water sites, while infection prevalence remained high beyond 1 km distance, suggesting that water contact interventions should be implemented at a smaller spatial scale than

MDA. With detailed data representing diverse geographies in Uganda and high predictive model performance, our findings have high internal validity. To further assess the generalisability of our findings, our methods should be applied to data from different countries and different schistosome species.

This study highlights the complexity of human exposure and the need for guidelines to consider water contact when determining the frequency of treatment. It also provides information for the WHO to create measurement tools to enable standardised data collection on water contact for use in country monitoring and control programmes.

## Methods

### Study setting, participant sampling, and assessments

We conducted a cross-sectional study in Uganda using baseline data from the SchistoTrack cohort[24]. Data were collected in January and February 2022 during the dry season when human water contact and snail abundance were high[49]. Thirty-eight diverse rural villages were sampled across three districts of Pakwach, Buliisa, and Mayuge. We sampled villages within a 3 km distance based on the village centre location reported by the local chairman of the shorelines of River Nile, Lake Albert and Lake Victoria (Fig. 2b–d). In all study districts, MDA with praziquantel has been carried out since the year 2003, targeting all children aged 5 + years. Thirteen or more rounds of treatment have been administered to date in each district. The most recent MDA was conducted over one year before this study. We randomly sampled a total of 1459 households, ~40 per village, from village registers or MDA records. Sample size was determined as reported in Anjorin et al.[31]. All households with at least one child and one adult residing in the village for at least six months of the year were eligible. Informed consent was obtained from all participants. Parents or guardians consented on behalf of children under 18 years of age. Adults provided written consent for children, and where possible, children also provided written consent. All children provided informed verbal assent. After obtaining consent, questionnaires were administered to obtain socio-demographics, biomedical variables, WASH and environmental variables, and water contact patterns. At the end of the household interview, one adult (aged 18 +) and one child (aged 5–17) per household were selected for clinical assessments by the household head. All participants were treated for schistosomiasis using praziquantel following clinical examinations, irrespective of schistosome infection status. Figure 1 depicts the participant flow resulting in an analytical sample of 2867 participants from 1444 households. A more detailed participant flow diagram is shown in Supplementary Fig. S13.

We also conducted village-level infrastructure and malacology surveys. For village-level infrastructure, we collected information on the number of functioning public latrines and public taps/boreholes and their respective GPS locations, as well as the GPS locations of all primary schools. To assess environmental risk, malacologists mapped all water sites in the study area, guided by the village chairman or a village health worker, and recorded GPS points, site type, and the observed presence of human faeces at all water sites within the village. Malacologists collected all living and recently dead snails at each water site for 30 min, using both scooping and handpicking, as well as checking floating vegetation. Snail infectivity was established by exposing snails to natural sunlight for a maximum of eight hours the day after collection and determining the number of snails shedding human schistosome cercaria by water site.

### Water contact outcome

Survey questions were administered to the household head and/or wife to elicit information on 11 different water contact activities conducted by household members, and their typical weekly frequency, duration, and time of day. The 11 activities were getting drinking water, washing clothes with/without soap, bathing with/without soap, washing jerry cans or household items, washing clothes, collecting papyrus, fishing, fish mongering, collecting shells, and swimming or playing. The choice of activities was informed by the literature and local collaborators to ensure all common water contact activities in the study area were captured. The typical duration per trip to waterbodies for each activity in the household was elicited using predefined duration categories (less than 30 min, 1, 2, 3, 4 h or more). For frequency, the typical number of trips per week in the household was collected. The typical time of day for each activity was elicited using the following categories: before sunrise (1-6 am), after sunrise (6-9 am), late morning (9 am-12 pm), midday (12-3 pm), late afternoon (3-5 pm), early evening (5-7 pm), after sunset (7-9 pm), late evening (9 pm-1 am).

Our main outcome was having any water contact which was coded as binary and defined as an individual engaging at least in one water contact activity per week. We chose having any water contact as the primary representation of exposure based on a recent meta-analysis which showed that having any water contact was associated with infection status[3]. Secondary outcomes included the type of water contact activity and the frequency and duration of water contact. For activity types, we grouped water contact into three distinct categories: domestic, occupational, and recreational water contact. Domestic water contact was defined as engaging in getting drinking water, washing clothes with/without soap, bathing with/without soap, washing jerry cans or household items, or washing clothes. Occupational water contact was defined as engaging in collecting papyrus, fishing, fish mongering or collecting shells. Recreational water contact was defined as swimming or playing. To obtain individual-level water contact frequency, we summed the self-reported weekly number of water contacts across all activities for each individual. The duration was obtained by multiplying the weekly frequency of each activity with its duration and summing durations across all activities for each individual.

For the water contact model in Fig. 9b, an additional 82 granular water contact variables indicating the frequency and duration for each of the 11 water contact activities, the time of day of each activity, and whether an activity was conducted at a high-risk time (10 am- 3 pm), were included. We also included variables for the frequency and duration of domestic, recreational, and domestic activities and variables indicating whether anyone in the household conducted any of the 11 water contact activities.

### *S. mansoni* infection outcome

Kato-Katz stool microscopy and point-of-care cathodic antigen (POC-CCA) diagnoses were completed for schistosomiasis as described in Anjorin et al.[31]. We prepared duplicate, thick smear slides from a single, stool sample produced on the morning of the slide preparations. Senior technicians re-read a random sample of 10% of all slides for quality control. Readings were converted to eggs per gram (EPG) of stool. Infection was defined as >0 EPG. In accordance with the current WHO treatment guideline[4], heavy infection intensity was defined as ≥400 EPG. We chose infection status (based on Kato-Katz diagnosis) as the primary infection measure as infection status is relevant for transmission, and prevalence has been used to monitor reductions in transmission[50] and identify persistent hotspots[51,52]. POC-CCA was not chosen as the primary infection measure due to its lack of specificity[53] and undemonstrated relevance for transmission[54]. POC-CCA was only used in sensitivity analyses to demonstrate the robustness of main models in the potential scenario of missed light infection intensities.

### Individual, household, and village human-environmental variables

Detailed definitions of all variables are provided in Supplementary Table S1. We curated a set of 33 candidate variables for our main analysis and aimed at comprehensively covering key human-environmental determinants of water contact and infection (27 variables for the water contact models and six additional water contact

variables for the infection models). At the individual level, we included self-reported age, gender, occupation, and highest education level completed as covariates, based on their relevance for water contact and infection[13,21,55,56]. Age was coded to the nearest year and was a continuous variable, as grouping age into categories could lead to information loss[57]. We generated an age$^2$ term to model nonlinear associations of exposure and infection.

Household-level variables were as follows. To account for the influence of socio-economic status, we generated a home quality score variable as described in Chami et al[5]. WASH access may influence water contact by crowding out open water contact or may affect infection via reducing faecal contamination and thus environmental risk[3,12]. We generated sanitation variables indicating the Euclidean distance from the household to the closest public latrine and the presence of a household home latrine. Having a flush or pour-flush toilet, a covered pit latrine with/without privacy, or a composting toilet was considered as having a home latrine. Households who reported open defecation or having only bucket latrines were considered as not having a home latrine, as per WHO-UNICEF Joint Monitoring Committee 2017 methodology guideline definitions[58]. We also generated water infrastructure variables indicating the Euclidean distance to the closest tap/borehole, household water purification method used (if any), the presence of a handwashing facility, and whether the household used public taps/boreholes as their primary source of drinking water. Overall, 52.8% of households (95% CI 51.0–54.6%) used public taps/boreholes as their primary source of drinking water. We also generated variables for the number of latrines, and the number of taps/boreholes per village.

At the village level, we recorded a wide range of characteristics for water sites, of which some were combined to create distance metrics. The types of water sites (river, marsh, beach, swamp, pond) in each village were recorded by malacologists. The number of water sites per village was calculated from water sites mapped by malacologists. To measure faecal contamination, malacologists observed at all sampled water sites whether there was human stool present. Past research has shown the existence of lake-gradients in infection[21,45]. We, therefore, generated Euclidean distance variables to assess whether these gradients were explained by variations in water contact behaviour across lake distance. Variables were defined as follows. Household lake distance was the Euclidean distance of the household GPS location to the closest water site mapped by malacologists. Village lake distance was computed by assigning households to the village centre, reported by the local chairman and calculating the Euclidean distance to the closest water site. For school lake distance, all children attending primary school were assigned to the school closest to the household. We then calculated the Euclidean distance from the school to the closest water site. Distance to the type of water site closest to the household was generated using information on the type of site collected by malacologists.

We collected Information about intermediate snail hosts and infection prevalence also was incorporated at the village level. The snail variables included were the number of water sites with infected snails per village, number of snails within 1 km of the household, number of snails per village, number of infected snails per village, presence of any infected snails in the village, distance to closest water site with infected snails, and whether the closest infected water site from the household was within the village. Village-level prevalence was defined as per WHO guidelines on the control and elimination of schistosomiasis with 10–49% prevalence by Kato-Katz corresponding to moderate and ≥50% prevalence corresponding to high endemicity[4]. There were no low-prevalence (≤10%) villages in our study.

## Comparison of self-reported water contact with direct water contact observations

Direct water contact observations were conducted in a subset of 12 villages concurrently with the collection of self-reported water contact

data. All water contacts at two water sites per village were recorded by trained local village health team members who were trusted by the communities and knew all individuals within their village. Contact was observed between 6 am-6 pm for two weeks, and each visitation of the water site, defined as having at least one body part immersed, was recorded as a separate water contact event. Age group, as defined per recent WHO guidelines on schistosomiasis control[4], as well as gender, activity, bodily immersion, time, and duration, were recorded. By comparing the age distribution in direct observation data for participants aged 5 + with the age distribution of water contact in self-reported data from the same 12 villages, we assessed the extent to which survey data were representative of the age composition in the observation data.

## Geospatial data

Waterbody boundaries shown in Fig. 2 were from a public shapefile of waterbodies in Uganda, made available online under a Creative Commons Attribution (Non-commercial 4.0 International) license by the World Resources Institute[59].

## Statistical analysis

All statistical analyses were conducted in R version 4.1.0. To describe the univariate non-linear trends of exposure and infection over age, we used GAMs from the mgcv package[60]. The number of knots was separately chosen for each GAM based on the mgcv diagnostic function 'k.check'. The GAM results informed our subsequent modelling of age in the regression models to predict water contact and infection, where we used age and age$^2$ terms to account for nonlinearities. We also modelled trends in water contact over distance to the nearest water site using GAMs and obtained distance gradients from these GAMs by computing the average slope over household distances between 0-1 km. We refrained from reporting distance gradients > 1 km household distance, as only 4.8% of participants (137/2867) lived in households > 1km from a water site. We used Spearman (ρ) correlation coefficients to assess correlations between water contact and infection status across participants. We also assessed pairwise correlations of these variables within households. All $t$ tests conducted in this paper were two-sided. To select predictors of water contact and infection status, respectively, the same variable selection procedure was applied separately using the same list of input covariates except for six additional exposure variables, which were only included in the infection status models. We included a district-level fixed effect in every model to account for unobserved district-level differences in water contact and infection. To assess the relevance of snail variables for exposure and infection as well as the relevance of more granular exposure variables for predicting infection, we used BVS to select additional predictors from the sets of seven snail variables and 82 granular exposure variables, respectively, while constraining all previously selected variables to remain included. This way, we tested whether adding snail variables or more granular exposure variables improved predictions compared to the models with our main sets of predictor variables. We implemented BVS using the Bayesian Adaptive Sampling (BAS) package in R[61]. The outputs of this analysis were a ranking of the best-fitting models with associated posterior probabilities for each model being the best model, as well as marginal inclusion probabilities bounded between 0 and 1 for each variable, indicating the importance of including each of the predictors in our water contact and infection models, respectively. The large number of candidate variables yielded a prohibitively large model space ($2^{27}$ possible models for water contact). Therefore, we used Markov Chain Monte Carlo (MCMC) methods implemented within the BAS package to sample models based on their posterior probabilities and avoid enumerating all possible models[62]. We ran $2*10^7$ MCMC iterations, a number deemed sufficient based on the in-built MCMC diagnostics of the BAS package. The prior model distribution, was set to a uniform distribution which assigned equal

probabilities to all models. For the distributions on the regression coefficients, we used Jeffreys-Zellner-Siow-priors, which have been shown to have desirable mathematical properties and which provide more consistent results under both the null and the alternative model than alternative priors such as hyper-g priors or Empirical Bayes priors and are also more consistent than non-Bayesian methods such as the Akaike Information Criterion[63–65]. To build the final exposure and infection models, we selected all variables with inclusion probabilities of pr ≥0.5 in the BVS output, which yielded the so-called median probability model[66]. The choice of pr ≥ 0.5 was motivated by research showing that the median probability model, not the highest probability model, often yields the best predictive performance[62]. As an alternative to BVS, we used LRTs with a cut-off of $p < 0.05$. For LRTs we compared a model with just one variable at a time against an empty model with village-level fixed effects only to select predictors of water contact and infection. A comparison of the variable sets selected using Bayesian variable selection and LRTs is shown in Supplementary Table S7. For models shown in Fig. 9b, c, we included the additional snail variables and water contact variables, described above (model results shown in Supplementary Figs. S15, 16).

After variable selection, we used generalised linear models with a binomial family and a log-link function to predict whether an individual had any water contact ($n = 2867$). For all regression models, we clustered standard errors at the household level using the 'sandwich' package in R[67] to account for our sampling design where households were randomly selected, but not individuals. We considered the use of both random and fixed effects models to account for unobserved household and village-level confounders. Using a Durbin–Wu–Hausman test[68], we compared the consistency of village-level random effects models against an alternative model using district-fixed effects. The test statistic from the Durbin–Wu–Hausman test comparing the water contact random effects model with the respective fixed effect model provided strong evidence ($\chi^2 = 137.46$, p-value < 0.01) that random effects models would have provided inconsistent estimates. We also ran the Durbin–Wu–Hausman test for the infection status model and found that a random effects model would have provided consistent estimates ($\chi^2 = 3.6994$, p-value = 0.99). Yet, when we analysed model residuals from the fixed-effects infection status model, we found that the intra-class clustering coefficient (ICC) was 0.017, indicating that there was very little unaccounted household-level variation. Therefore, we believe that the use of district-level fixed effects was justifiable both for modelling water contact and infection status.

To identify which variables predicted a specific type of water contact, we used the main set of selected variables from the water contact model to predict domestic, occupational, and recreational water contact, respectively, in separate logistic regression models. In these models, we removed all participants engaging in multiple types of water contact activities to generate mutually exclusive categories. This removal resulted in 6.4% of individuals (183/2867) being excluded. To adequately model the overdispersion of water contact frequency and duration, we used negative binomial models to predict frequency and duration using all selected predictors of exposure. For all negative binomial models, we tested whether they adequately modelled zero inflation using the 'check_zeroinflation' function from the performance package in R[69]. For infection, we used the selected infection variables from Bayesian variable selection in logistic regression models to predict infection status as well as heavy infection ($n = 2867$). We modelled the frequency and duration of water contact as well as infection intensity using negative binomial regression models. Across all models, we computed generalised variance inflation factors (gVIFs) to diagnose multicollinearity[70] but did not remove any variables as none had gVIFs > sqrt(10). Ten-fold cross-validation was used to assess the predictive capacity (auROC) of the statistical models with binary outcomes.

## Reporting summary

Further information on research design is available in the Nature Portfolio Reporting Summary linked to this article.

## Data availability

The raw data are protected and are not available due to data privacy laws. Processed metadata for all variables has been provided in the manuscript and supplement. Demonstration data with the same structure and number of variables as the restricted data, which enables rerunning the analysis pipeline, is available at: https://doi.org/10.25446/oxford.26977897.

## Code availability

Code to rerun the entire analysis pipeline, the main Figs. (3–9) as well as all Supplementary Figs. and tables are available on Figshare at https://doi.org/10.25446/oxford.26977915.

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

## Acknowledgements

We are thankful for the involvement of our study participants and local communities. We thank all field teams, including malacologists, surveyors, nurses, and laboratory technicians involved in the 2022 SchistoTrack baseline data collection, especially Benjamin Tinkitina, who led the household survey team. The support of the Uganda Ministry of Health, local district leaders, focal health workers, and village health teams was crucial for building partnerships and continued trust with study communities.

## Author contributions

Conceptualisation: F.R. and G.F.C. Data curation: F.R., N.B.K., A.M.B., F.B., B.N., and G.F.C. Formal analysis: F.R. Funding acquisition: G.F.C. Investigation: F.R. and G.F.C. Methodology: F.R. and G.F.C. Project administration: N.B.K. and G.F.C. Resources: N.B.K. and G.F.C. Software: G.F.C. Supervision: G.F.C. Validation: F.R. Visualisation: F.R. Writing – original draft: F.R. Writing – review & editing: F.R., N.B.K., A.M.B., F.B., B.N., and G.F.C.

## Funding

A DPhil scholarship was awarded from the Nuffield Department of Population Health (NDPH) to F.R. Grants from the Wellcome Trust Institutional Strategic Support Fund (204826/Z/16/Z), NDPH Pump Priming Fund, John Fell Fund, Robertson Foundation Fellowship, and UKRI EPSRC Award (EP/X021793/1) were awarded to G.F.C.

## Competing interests

The authors declare no competing interests.

## Ethics approval

Data collection and use were reviewed and approved by the Oxford Tropical Research Ethics Committee (OxTREC 509-21), Vector Control Division Research Ethics Committee of the Uganda Ministry of Health (VCDREC146), and Uganda National Council of Science and Technology (UNCST HS 1664ES).
