## [Peer Review file · Nature Communications]

Current water contact and *Schistosoma mansoni* infection have distinct determinants: a data-driven population-based study in rural Uganda

Corresponding Author: Professor Goylette Chami

Version 0:

Reviewer comments:

Reviewer #1

(Remarks to the Author)

The paper by Reitzug et al. provides a state-of-the-art, data-driven analysis of water contact rates and associated transmission risk for schistosomiasis for 2,867 individuals from 1,459 households in 38 fishing rural villages with 3km distance from either River Nile, Lake Albert, and Lake Victoria where schistosomiasis is endemic. The authors present as a detailed breakout for both contact rate and infection rates by age, gender, type (residential, occupation, recreational, etc.) and their association with a number of covariates (socio-economic, demographics, environmental and WASH), analyzed by using generalized, additive, mixed effects models (GAMMs), logistic regression (when classifying one or more contacts as 1, zero: no contacts; likewise for infection) and using Bayesian variable selection to identify the most important driving variables. Most of the data were gathered through questionnaires administered to randomly selected householders from the above-mentioned villages, or through direct observation, whereas infection levels in the study population were derived through the Kato-Katz technique (and also through Point of Care Circulating Cathodic Antigen).

The authors present a massive amount of information on the analyses and the results and I commend them for such an incredible sampling effort and for a very thorough analysis that allows them to gather a much better understanding of exposures and infection in their multiple dimensions. Their study casts light on very fine nuances of contact rate with water and, in some cases, it challenges the conventional knowledge and/or good sense expectations. The paper is informative and well written in general aside some minor point. In addition to a lot of very useful information, there are two major results according to my humble understanding of the field.

First, this study clearly shows that age-specific infection curve peaks at a much younger age (~15 years yo) than the peak of the age-specific contact rates with water (which peaks at 30 yo). This is a very interesting result that provides indirect support to the hypothesis of a slowly mounting immune response with age.

Second, and perhaps the most surprising result, there is basically no correlation at the individual level between multiple metrics of water contacts and infection levels in the studied population – which is mind-blowing, as it challenges the conventional knowledge that the other the contact rate with potentially contaminated waters, the higher the probability of infection and the infection intensity.

As a third point, I commend the paper for casting light on the multiple dimensions water contact can be characterized.

For the massive sampling effort, the breadth and depth of the analyses and the novelty of the results, I believe that this paper deserves to be published on Nature Communication. Before accepting the paper, anyway, there is a few issues that the authors may have to address.

First, the authors should consider the possibility of reporting at the beginning of the Results section (which is before Methods in journal's format) more information about what types of contact rates have been considered and how the information has been gathered. For instance, when I read the paper the first time, it was unclear to me whether householders were asked to report the average contact rate in the last week, in the last two weeks or longer. I am aware that this information is provided at line 496 and 502 in the Methods section. Anyway, as this paper is entirely built upon water contact rates, I think that it could be useful to provide more information upfront. The authors could do it by considering to moving at the beginning of the Results section the first paragraph (or so) of the Exposure section (lines 495 to 507) or, in alternative, at the end of the introduction/background, possibly including also the first paragraph (or a synthesis) of the Methods section, the one on the study area, the number of householders interviewed, etc.

Also, sometimes I got lost a bit in the wealth of information presented in the results and discussion section. Obviously, this wealth of information is a plus of the study, but I was wondering whether the discussion could be partially reorganized to outline what I personally believe are the three most important take-home messages of this paper namely:

1. While we usually tend to think of water contact rate as a single parameter in disease dynamics models of schistosomiasis, contact rate is actually a complex metric with multiple dimensions (by age, gender, time of the day, type of contacts, etc.)
2. Age-specific infection curve peaks 15 years before than the peak of the age-specific contact rate with water
3. There is no correlation between water contacts and infections at the individual level

The authors are good at giving justice to point (1). As for point (2) they rightly emphasize that their results do not support the hypothesis that the peak in the age-specific infection prevalence corresponds to the peak in water contact rate. And yet, I feel the authors should attempt to present possible explanations of (2) and (3).

For instance, the authors state that <the role of different exposure measures [...] and acquired immunity [...] needs to be explored in future studies> (lines 332-333), but the statement comes out of the blue, as this is the first time (unless I have missed it) they bring up immunity in the attempt to explain infection outcomes. I think that before making that statement the authors could acknowledge the literature showing (or speculating) that it takes time for the immune response to build up and that their result (the mismatch between age-specific peaks in water contact rates and infection) can be considered supportive evidence that this is the case.

Along the same line, the authors could provide a more comprehensive discussion of alternative explanations of the findings (for peaks' mismatch and/or for lack of correlation between exposure and infection), including the following among the other:

- Not all water contacts are alike: for instance, fishermen are most active in the late evening when snails are not expected to shed and probably risk is minimal. This is true also for the early morning water contacts of girls, for instance (by the way, if this is the case, maybe the authors could try to correlate infections only with contact rates occurring in the central hours of the day, when the transmission risk is supposedly the highest. Or, they could use as covariate a weighted sum of frequency or durations of water contacts, where weights go from zero, in the night, to 1 at the pick of infected snails' shedding rate – just an idea).

- Difference among the three regions (and two lakes and one river locations), although I assume that authors' mixed effect models used the three locations as fixed factors with villages (random factor) nested in location, and the individuals in study population nested in villages, or something like that – I do not think that this is specified in Methods, unless it is in the supplementary information, so the authors could clarify it)

- Seasonal differences between water contact rates and snail population peak and seasonal peak of the transmission risk.

For instance, Andrus et al. (PLOS NTD, 2023) found seasonal differences in snail prevalence of infection in Lake Victoria in Uganda (with slightly higher prevalence in the wet season). Here the surveys have been conducted during the dry season in winter where, according to a recent paper by Civitello, *B. sudanica* might estivate under harsh climatic conditions. In the hypothetical case water contact rate changes between the dry and wet season, it is possible that the frequency and duration of water contact rate that the householders reported in Jan-Feb might not correspond to the time of the year when infection is most likely to occur (just an hypothesis, the authors are better equipped to assess whether it is even worth considering).

- The authors comments that, at least in the case of children's water contact rate, householders' report might be biased by uncertainty and unreported events in which children steps into contaminated waters.

- Last but not least, the authors my acknowledge that there might be additional drivers that were not accounted for in this study. For instance, village level prevalence of infection >50% is a significant covariate for individual level infection (line 357): so, if schistosomiasis is widely circulating in a village, it is more likely that the an individual children is infected, which is either trivial – if village level prevalence is high, it is obvious that there is a higher probability that any random children is infected by definition, or illuminating: where prevalence is high, it is more likely that the parasites will contaminate the water when people defecate close to the water body, which means that contact with highly contaminated water occurs. In a completely different pathogenic landscape, Wood et al. (PNAS 2019) and Jones et al. (PLOS NTD) 2021 showed that there are fine-scale features of the environment (in that setting the amount of floating vegetation, the shape of the water access points) that contribute to explain village specific transmission rates. We do not know what features might be at play in the authors' study site, but there might other ecological and environmental drivers not accounted in this study that might help to understand when water contact rate might lead to the chance of infection.

I report below a series of more or less minor comments.

line 57-58: consider to break the long sentence by putting a period after households (line 57), and starting the new sentence with "In addition, there are concerns...")

line 93: "water contact data were collected." This is where I think that, if possible and according to journal guidelines, it would increase clarity if you moved here the first paragraph of the "Exposure" section.

Line 107: <current water contacts>. Why current? The water contacts of the week? As I mentioned above, current water contacts in the dry season might when snails prevalence is low might be less risky than when water contacts occurs in the wet season (which might help to explain the lack of correlation between water contacts and infection)

Line 110: <high intensity water contact>, not sure what the authors mean with high intensity? Long duration? More than 20h/week? Or entire body vs only feet? Who are these "few individuals" fishermen, boys, or distributed more or less randomly among the various demographics?

Line 129: I am certainly a bit picky here, but how do you have a water contact that does not occur around or close to a water body? Where do the remaining 20% of the water contact occur?

line 134: I suggest starting the sentence "We used generalized additive models" in a new line. Also the authors might consider to add "mixed-effect" to generalized additive model (GAMMs).

Line 135: maybe I am wrong here, but somehow, I tend to associate “prevalence” to infection, so it sounds weird to me to talk about “prevalence of water contact” although semantically correct (unless the author specifies: <... (i.e., fraction of the studied population that has one or more water contacts per week)>, or something like that). This is true also for line 152

Line 147: <The duration of water contact...>: I think that here the authors meant the frequency, not the duration.

Line 160-162: if possible, add a reference to support the statement on the peak time for cercarial shedding rate

Line 174 and following. I got a bit confused here, as I think (but I might be wrong) that BVS is used in association with the GAMMs, but it is not explicitly said so in this paragraph, so it seems that BVS is used to thin the number of variables before running any model, which I don't think it is the case (unless I am plain wrong).

Line 190 and following: is the main exposure, logistic model where the dependent variable is whether an individual had one or more water contacts?

Line 198: what type of contamination? Cercarial contamination, chemical contamination (soap foam, plastic pollution), or simply turbidity/murky waters?

Lines 247-248. As this paragraph is about infection, I would turn this sentence reporting first the information on number of variables for infection status (7), in contrast to the 10 variables used as predictor of water contact.

Line 272: this sentence on endogeneity is a bit obscure to me, can the authors help the reader (or just me) to understand what is this about

Line 333: as I mentioned, here acquired immunity comes out of the blue, with no context why it is important (i.e., because it takes time to mount) and what are the implications (see comments above)

Lines 336-337: <After adjustments for covariates such as age> (and what else?), the authors might consider to use: <After controlling for age, xxx and yyy, females were more likely ...> - or something like that.

Line 340: after <... than males> consider to add: in agreement with [REF] >

Lines 347-349: I do not understand what the authors mean with: <... by independent factors that drive gender difference in exposure, but not gender itself>.

Line 345: It seems to me incorrect to talk about <key risk factors for water contact>, I consider them <driving factors>, not risks, as water contact is not a risk per se, it is essential for the life of these populations, although I agree that any water contact in parasite contaminated waters accrues the risk to acquire further infections.

Line 357: what does it mean “having primary-school education”? “to have a primary school degree” or “to attend school”?

Line 360-361: personally, I consider very positive that exposure reduction interventions have been targeted at school age children as these are the age classed with the highest prevalence of infection, so, the most vulnerable one, regardless of their actual contact rate. I think that the goal of MDA is to reduce morbidity, not necessarily to target the most exposed classes.

Line 361-362: as this study does not solve the conundrum of a lack of correlation between water contact and infection (so, we do not understand how people get infected), I think that this is a bit of an overstatement and my personal suggestion is to remove or down tone this sentence.

Line 406: a 3.4% reduction every 100 meters means a nearly ~30% reduction the fraction of the study population having contact with water in 1km, and a ~65% reduction in 3 km, which is remarkable and worth to emphasize.

Line 408: is 1.9% the reduction in infection prevalence every 100 meters (within 1km from the shoreline – just half of the reduction rate in pr) or in 1 km? in the former case (as I think), I think that the sentence is a bit misleading. In the latter case, I agree that, although significant, it would be truly a negligible reduction, which means that everything in the first km has the same chance of being infected.

Lines 593 on: kudos to the authors for the sophisticated statistical analyses (and please, clarify whether the three regions are used as fixed effects and if villages (random factors) are clustered within the region and the study population clustered within villages?)

A last consideration: I do understand that the confidentiality of the data prevents to make them public. I still think that the scientific community (and especially graduate students in training) would learn a lot from the R scripts the authors used (maybe we a randomized subset of data, where the three studies locations and the village names are anonymized) – I would encourage the authors to make their R script public, either as supplementary information or on GitHub.

I trust the authors will be able to address these points, and I apologize in advance for the cases in which I have misinterpreted authors' intentions and what was clearly reported in the manuscript.

Reviewer #2

(Remarks to the Author)

This is a nicely written paper, summarising a very comprehensive and carefully executed study. The authors are to be

commended. I only have a few comments:

1. The novel implications/recommendations for improved schistosomiasis control strategies arising from this research need to be more explicitly explained. I would suggest adding this as a distinct section near the end of the paper, and summarised in a sentence or two in the abstract. In doing so, please explain to what degree these recommendations may be specific to the case study sites in Uganda versus universal.
2. The assumption seems to have been made that all of the studied water bodies are of equal risk in terms of cercarial contamination. Is there any evidence to support this?
3. In terms of fishing, does this tend to occur near the shoreline, or out in the lakes? Snail and cercariae levels would be expected to be higher near the shoreline.
4. In terms of washing, do all people use soap when doing so? If not, it might be interesting to compare infection levels of those using soap and not.
5. The way that the two sentences on lines 147-150 are worded makes them seem directly contradictory - 'The duration of water contact was higher for females than males...Females had a lower duration of water contact than males...' Suggest rewording. Check carefully for other contradictions elsewhere in the paper.
6. Related to my first comment, it would be a shame if a policymaker read this paper and came away with the conclusion that addressing water contact is not important, since the determinants of exposure are not the same as the determinants of infection. Ultimately all infections were caused by exposure at some point in the lifetime of the infected person via some kind of water contact activities, so this should be emphasised.

Reviewer #3

(Remarks to the Author)

General comments

The authors investigated potential correlation between self-reported frequency and timing of water contact activities (dependent variable) and a host of demographic, socio-economic, WASH, and geographic variables (independent variables) among ~1400 households across 38 rural fishing villages in Uganda. The same independent variables were used to assess a potential correlation with infection as measured by Kato-Katz (or an antigen test). Presence of snails and infected snails were considered as additional predictors. Self-reported data on water contact were further compared to directly observed behaviour in 12 villages. The study is a very valuable effort to better understand the link between WASH variables, water exposure and (age patterns) in infection. Overall, I'm positive about the manuscript, but think it would benefit from some additional clarifications, reformatting of certain figures, rephrasing of key messages, and additional analysis of infection intensity data.

Introduction

Regarding: "For estimating the force-of-infection, current mathematical models assume age-specific prevalence to be proportional to age-specific trends in current water contact [22] and do not always account for individuals who remain infected for years without current exposure due to the long lifespan of the schistosome parasite [23]."

The first part of the statement about reliance of models on age patterns in water contact may be true for some models, but not all. In some models (e.g., Imperial College London), the age patterns in exposure are estimated from the observed patterns in intensity and prevalence of infection by age. Regarding the second part of the statement about not always accounting for the long lifespan of schistosomes: I'm not aware of any mathematical models that do not account for this. The lifespans of the adult worm pairs in the human host are always explicitly simulated as this aspect dictates the population dynamics of SCH for a large part.

Of note is that all existing mathematical models SCH assume that acquired immunity, if any, does not affect the probability of parasite establishment on exposure. As such, all model-estimated and predicted age patterns in exposure and infections levels are conditioned on this assumption.

Methods

Regarding: "This study was designed to detect a minimum effect size of 8% with an unevenly exposed population and a household design effect of 1.136 at 97.5% power."

Please clarify what the expected minimum effect size is referring to. A difference in water exposure and/or infection prevalence/intensity? Between which groups? And what was the (seemingly small) household design effect of 1.136 based on? Was this design effect intended to capture correlation of observation within households and/or within villages?

Please clarify whether single or duplicate Kato-Katz was done (or something more elaborate, e.g. multiple days).

Was there a particular need to round age to the nearest year? This is emphasised in both the methods section and the figures. Was there a particular reason to explore age polynomials up to the 4th order? E.g., did you expect two peaks in infection or exposure levels across the range of sampled ages? Also, this information would be expected in the statistical analysis

section and not the data collection section. Moreover, later on in the “statistical analysis”, the use of generalised additive models (GAMs) is mentioned, with explicit mention of the use of knots to model non-linear age patterns. This seems to be a different approach than the use of simple polynomial terms. Please clarify.

The authors used Pearson correlation coefficients to assess correlations between water contact and infection status across participants. First off, this implies that these variables are assumed to be correlated in a linear fashion, which is arguably doubtful (e.g., why not use Spearman correlation).

When describing the water contact model, the authors state that “For all regression models, we clustered standard errors at the household level to account for our sampling design where households were randomly selected, but not individuals.” How was this handled in the models for infection? There, I would expect additional clustering of residuals at the village level, as infections in the same villages are connected via their shared environmental reservoir of infection.

Data availability: the authors state that the “Data is not publicly available due to data protection and ethics restrictions related to the ongoing nature of the SchistoTrack Cohort and easily identifiable nature of the participants. Code can be shared upon reasonable request to the study authors.” The offer to share only code on reasonable request is not in the spirit of open science. Code can be stored separately and can be easily shared via open-source repositories like GitHub or GitLab. Also, I don’t see why the data cannot be made publicly available. Why can’t the data be pseudo-anonimised, removing personal identifying data and geolocation data and replacing individual, household and village identifiers with codes to which the authors themselves only have the codebook to identify individuals? I think making both the code and data available is important, in particular given the data richness and the many ways in which the research question could have been answered with different statistical methods and approaches.

Why not also model intensity of infection and only presence of infection? Intensity data contains much more information than prevalence data.

Results

Up to line 171, the authors describe a lot of summary statistics for many different variables. At this point during the reading, I was not clear on whether these variables had already been selected or not with the Bayesian variables selection, or is this a description of everything that was measured? Only after line 171, the reader gets to Figure 6, which provides an overview of the variables that made it through the selection procedure. It would be helpful if the reader can be guided a bit more through what

For all figures with probabilities or percentages on an axis: please use increments of 20% instead of 25% for tick marks. Interpreting data points or estimates half-way a 25% tick mark is very awkward. And please consider adding grid lines at 10% increments. These adjustments would significantly improve the readability of figures.

“Our study population was characterised by a high prevalence of self-reported water contact with open freshwater bodies (46.7%, ...)”. Please clarify what is meant here by water contact. Does this mean that 53.5% of the participants said they never has any water contact? Or perhaps water contact at some frequency less than XX? Something else? Please help the reader understand what you quantified here. The next sentence refers to “current water contact”, but it is also not clear what is meant by that without going to the questionnaire. Although the authors state that “Detailed variable definitions and characteristics of the 2867 participants are presented in Tables S1-2”, I would expect the definition of “water exposure” as one of the main outcomes to be explained and described in the main text and not in a supplement.

There seems to be a numerical discrepancy between the two following statements made by the authors:

- Line 105-106: “Our study population was characterised by a high prevalence of self-reported water contact with open freshwater bodies (46.7%, ...)”

- Line 129-130: “Water contact was highly concentrated around waterbodies as 80% of all participants with water contact (and 64.3% of all participants) lived in households within 0.34 km Euclidean distance of a water site”

Given the first statement, the 80% in the second statement would translate to $80\% \times 46.7\% = \sim 37\%$ “of all participants”, whereas it says “64.3% of all participants”.

“3.4% absolute reduction” consider “reduction of 3.4%-points” or “...3.4 percentage points”.

Figure 7: because the estimated odds ratios are plotted on a linear scale, most of the graphics are almost impossible to interpret. Many are very close to 1.0 or have extremely narrow confidence bounds. Sometimes this is also due to the chosen scale at which predictors are expressed (e.g., age, which could also have been expressed as “decades”, which is just as arbitrary as “years”, but might yield more legible estimates). A first step towards addressing this would be to plot the estimates on a logarithmic scale with equally spaced ticks at OR = 0.5, 1, 2, 4, etc.

The interpretation of the coefficient for age2 seems off: “The age2 term (odds ratio (OR) 0.9978; 95% CI 0.9975 – 0.9982), indicated significant nonlinearity but only a 0.0022% decrease with each one-year age increase.” This coefficient is the change in log-odds per unit change in age2, so not a “one-year increase”. As mentioned above, the scale of this coefficient is this small because of the choice of scale for age (and consequently the enormous scale for age2). More generally, it’s not very meaningful to try to interpret the coefficients for the linear and square age terms separately; it’s the set of the two together that explains the pattern. This reminds me, no mention is made of the “knots” (which are suggestive of the use of splines) that are mentioned in the methods section as a way to model the non-linear age patterns. The additional presence

of such a component in the regression model would make the interpretation of the individual coefficients for age and age² completely meaningless. If splines are indeed used, the age patterns are best described in a qualitative manner, with reference to Figure 4.

Why was village-level SCH prevalence dichotomised as a predictor? For age, you explicitly argue to use as a continuous predictor so as to not lose information.

Note for the discussion section: several of the independent variables in the regression analyses are measured with considerable uncertainty/sampling error. The larger this (random) measurement error in an independent variable, the more one underestimates its correlation with the outcome (dependent variable) due to regression towards the mean.

Regarding: "In terms of WASH, we found that in villages where all study households used a safe drinking water source (taps/boreholes), individuals had a 65% lower likelihood of having water contact compared to people living in villages where all study households used surface water (OR 0.35, 210 95% CI 0.24 – 0.52)."

This seems an unnecessary level of data aggregation for access to a safe drinking water source. Why was this association quantified at the village level and not the household level (e.g., using a 0/1 indicator for use of a safe drinking water source)? Unless there is a good reason (which the authors should then explain), such an unnecessary aggregation poses a risk of unintentionally introducing ecological fallacy into the analysis.

Discussion

Regarding: "We found that at-risk groups for water contact and infection differed in their observable characteristics suggesting that transmission/exposure and infection control interventions should not be targeted to a single or the same risk group."

This second part of this conclusion (underlined) is formulated in rather strong words. I had a very strong knee-jerk response of disagreement with this: of course, water exposure leads to acquisition of new worms, but potential regulation of parasite establishment by acquired immunity in humans may well explain the observed difference in age patterns; so why this strong statement about (not) targeting the same or different groups? But I then realised I'm not even sure what the authors mean here. What do the authors mean with "targeted to a single group or the same risk group"? This conclusion is also not substantiated further on in the discussion. Perhaps it would be helpful if the authors formulated their conclusion in terms of what they think "should" happen, rather than "should not". Also, I think the insight this study provides into a potential role of acquired immunity in the human host deserves to be mentioned in this first paragraph.

"We suggest that the effects of gender on exposure and infection are directly mediated by independent factors that drive gender differences in exposure but not gender itself." I'm not sure what the authors mean here. Please consider to rephrase. I'm also missing other potential explanations for the absence of an association between gender and infection: (1) infection is defined as positivity/negative whereas intensity of infection (eggs per gram faeces) can still vary between individuals; (2) can the authors really rule out that the (absence of) gender difference is explained by (absence of) gender differences in the type and location of water contact? See also my earlier comment about regression towards the mean with imperfectly measured predictors.

Other points

The typesetting of the document made the text rather hard to read: letters of the same words are often squashed together and/or irregularly spaced. For example, see the letter "i" (in general) and the word "infection" in this sentence:

Version 1:

Reviewer comments:

Reviewer #1

(Remarks to the Author)

The authors have carefully addressed reviewers' concern and provided a comprehensive, very detailed, point-by-point rebuttal to reviewers' criticism.

I am personally satisfied with the answers and the revised version of the ms, both as for my specific report and for the detailed and insightful analyses of the other two reviewers

Reviewer #2

(Remarks to the Author)

The authors have addressed my earlier comments satisfactorily.

Reviewer #3

(Remarks to the Author)

The authors have carefully addressed my earlier comments and the manuscript is much improved. I recommend for it to be published. I only have a few minor points:

1) Related to my earlier comment, about the now-revised sentence: "*For estimating the force-of-infection, current mathematical models infer age-specific water contact levels from infection trends over age. **As such, the models simply scale levels of water contact to be proportional to age-specific infection levels.***"

Although this revision improves on the previous version, the part in bold does not correctly capture what current mathematical models assume. Technically, math models assume that age-specific infection levels are a non-linear (not "proportional") function of the cumulative exposure (water contact) up to that age and the lifespan/survival of adult schistosomes within host. The associated statement on this aspect in the discussion is fine though: "*In terms of modelling, this study advocates for relaxing assumptions about a direct correspondence between current age-specific water contact rates and infection prevalence.*" Maybe the formulation in the introduction can borrow from the discussion.

2) In a few places in the results section, odds ratios are presented as "*X times higher likelihood...*". As likelihoods are probabilities, I suggest to rephrase these instances to "*X times higher **odds**...*".

REVIEWER #1

Reviewer 1, comment 1: The paper by Reitzug et al. provides a state-of-the-art, data-driven analysis of water contact rates and associated transmission risk for schistosomiasis for 2,867 individuals from 1,459 households in 38 fishing rural villages with 3km distance from either River Nile, Lake Albert, and Lake Victoria where schistosomiasis is endemic. The authors present as a detailed breakout for both contact rate and infection rates by age, gender, type (residential, occupation, recreational, etc.) and their association with a number of covariates (socio-economic, demographics, environmental and WASH), analyzed by using generalized, additive, mixed effects models (GAMMs), logistic regression (when classifying one or more contacts as 1, zero: no contacts; likewise for infection) and using Bayesian variable selection to identify the most important driving variables. Most of the data were gathered through questionnaires administered to randomly selected householders from the above-mentioned villages, or through direct observation, whereas infection levels in the study population were derived through the Kato-Katz technique (and also through Point of Care Circulating Cathodic Antigen).

The authors present a massive amount of information on the analyses and the results and I commend them for such an incredible sampling effort and for a very thorough analysis that allows them to gather a much better understanding of exposures and infection in their multiple dimensions. Their study casts light on very fine nuances of contact rate with water and, in some cases, it challenges the conventional knowledge and/or good sense expectations. The paper is informative and well written in general aside some minor point. In addition to a lot of very useful information, there are two major results according to my humble understanding of the field.

First, this study clearly shows that age-specific infection curve peaks at a much younger age (~15 years yo) than the peak of the age-specific contact rates with water (which peaks at 30 yo). This is a very interesting result that provides indirect support to the hypothesis of a slowly mounting immune response with age.

Second, and perhaps the most surprising result, there is basically no correlation at the individual level between multiple metrics of water contacts and infection levels in the studied population – which is mind-blowing, as it challenges the conventional knowledge that the other the contact rate with potentially contaminated waters, the higher the probability of infection and the infection intensity.

As a third point, I commend the paper for casting light on the multiple dimensions water contact can be characterized.

Response: Thank you for this direct and accurate summary of the study. We appreciate the suggestions later provided and have incorporated all suggested changes, which have greatly strengthened the manuscript. We also better highlight the three key results that you have emphasised. We provide a more extensive discussion around acquired immunity in paragraph three of the discussion section, as per your first point. Second, we highlight the lack of correlation between water contact and infection in the first discussion paragraph and also add new results showing that there is also only weak correlation between village-level infection prevalence and the proportion of individuals with water contact per village. Third, we incorporate your point about the multiple dimensions of water contact and highlight the

implications of this complexity for future research and control programmes in the eighth discussion paragraph. This change has greatly strengthened the potential impact of our study.

Reviewer 1, comment 2: For the massive sampling effort, the breadth and depth of the analyses and the novelty of the results, I believe that this paper deserves to be published on Nature Communication. Before accepting the paper, anyway, there is a few issues that the authors may have to address. First, the authors should consider the possibility of reporting at the beginning of the Results section (which is before Methods in journal's format) more information about what types of contact rates have been considered and how the information has been gathered. For instance, when I read the paper the first time, it was unclear to me whether householders were asked to report the average contact rate in the last week, in the last two weeks or longer. I am aware that this information is provided at line 496 and 502 in the Methods section. Anyway, as this paper is entirely built upon water contact rates, I think that it could be useful to provide more information upfront. The authors could do it by considering to moving at the beginning of the Results section the first paragraph (or so) of the Exposure section (lines 495 to 507) or, in alternative, at the end of the introduction/background, possibly including also the first paragraph (or a synthesis) of the Methods section, the one on the study area, the number of householders interviewed, etc.

Response: Thank you for this suggestion. It very much helped us to navigate a tricky aspect of the journal format which is that methods are only provided at the end of the manuscript. We now added a short description of the types of water contact, their data collected, and coding in the last paragraph of the introduction. The water contact reported was asked to represent the 'usual' or typical patterns of water contact in a week and can be interpreted as a self-reported average.

Reviewer 1, comment 3: Also, sometimes I got lost a bit is the wealth of information presented in the results and discussion section. Obviously, this wealth of information is a plus of the study, but I was wondering whether the discussion could be partially reorganized to outline what I personally believe are the three most important take-home messages of this paper namely:

1. While we usually tend to think of water contact rate as a single parameter in disease dynamics models of schistosomiasis, contact rate is actually a complex metric with multiple dimensions (by age, gender, time of the day, type of contacts, etc.)
 2. Age-specific infection curve peaks 15 years before than the peak of the age-specific contact rate with water
 3. There is no correlation between water contacts and infections at the individual level.
- The authors are good at giving justice to point (1). As for point (2) they rightly emphasize that their results do not support the hypothesis that the peak in the age-specific infection prevalence corresponds to the peak in water contact rate. And yet, I feel the authors should attempt to present possible explanations of (2) and (3). For instance, the authors state that (lines 332-333), but the statement comes out of the blue, as this is the first time (unless I have missed it) they bring up immunity in the attempt to explain infection outcomes. I think that before making that statement the authors could acknowledge the literature showing (or speculating) that it takes time for the immune response to build up and that their result (the mismatch between age-specific peaks in water contact rates and infection) can be considered supportive evidence that this is the case.

Response: Thank you for clearly noting the main contributions of this study. We have revised the discussion section in response to your comments. To address your first point, we now highlight the complexity of water contact throughout the discussion, especially in paragraph

four where we discuss gender differences in water contact, and in paragraph eight where we outline the implications of the complexity of water contact for research and control programmes. We now more clearly emphasise your second point as one of our major contributions and provide an extensive discussion in paragraph three of the discussion section. We thank you for your useful comment in point three and have provided a more comprehensive discussion around the lack of correlation between water contact and infection, including the role of acquired immunity in paragraph three of the discussion section.

Reviewer 1, comment 4: Along the same line, the authors could provide a more comprehensive discussion of alternative explanations of the findings (for peaks' mismatch and/or for lack of correlation between exposure and infection), including the following among the other:

- Not all water contacts are alike: for instance, fishermen are most active in the late evening when snails are not expected to shed and probably risk is minimal. This is true also for the early morning water contacts of girls, for instance (by the way, if this is the case, maybe the authors could try to correlate infections only with contact rates occurring in the central hours of the day, when the transmission risk is supposedly the highest. Or, they could use as covariate a weighted sum of frequency or durations of water contacts, where weights go from zero, in the night, to 1 at the pick of infected snails' shedding rate – just an idea).
- Difference among the three regions (and two lakes and one river locations), although I assume that authors' mixed effect models used the three locations as factors with villages (random factor) nested in location, and the individuals in study population nested in villages, or something like that – I do not think that this is specified in Methods, unless it is in the supplementary information, so the authors could clarify it)
- Seasonal differences between water contact rates and snail population peak and seasonal peak of the transmission risk. For instance, Andrus et al. (PLOS NTD, 2023) found seasonal differences in snail prevalence of infection in Lake Victoria in Uganda (with slightly higher prevalence in the wet season). Here the surveys have been conducted during the dry season in winter where, according to a recent paper by Civitello, *B. sudanica* might estivate under harsh climatic conditions. In the hypothetical case water contact rate changes between the dry and wet season, it is possible that the frequency and duration of water contact rate that the householders reported in Jan-Feb might not corresponds to the time of the year when infection is most likely to occurs (just an hypothesis, the authors are better equipped to assess whether it is even worth considering).
- The authors comments that, at least in the case of children's water contact rate, householders' report might be biased by uncertainty and unreported events in which children steps into contaminated waters.
- Last but not least, the authors my acknowledge that there might be additional drivers that were not accounted for in this study. For instance, village level prevalence of infection >50% is a significant covariate for individual level infection (line 357): so, if schistosomiasis is wildly circulating in a village, it is more likely that the an individual children is infected, which is either trivial – if village level prevalence is high, it is obvious that there is a higher probability that any random children is infected by definition, or illuminating: where prevalence is high, it is more likely that the parasites will contaminate the water when people defecate close to the water body, which means that contact with highly contaminated water occurs. In a completely different pathogenic landscape, Wood et al. (PNAS 2019) and Jones et al. (PLOS NTD) 2021 showed that there are fine-scale features of the environment (in that setting the amount of floating vegetation, the shape of the water access points) that contribute to explain village specific transmission rates. We do not know what features might be at play in the authors' study site, but there might other ecological and environmental drivers not

accounted in this study that might help to understand when water contact rate might lead to the chance of infection.

Response: Thank you for highlighting five important possible explanations for our reported disconnect between water contact and infection across age groups and the low individual-level water contact-infection correlations: 1) differences between demographic groups in the likelihood of water contact resulting in infection risk due to diurnal cercarial activity, 2) unaccounted regional differences in water contact patterns, 3) the potential that our study did not capture the peak transmission season, 4) underreporting of children's water contact, and 5) additional environmental (or other) factors not accounted for in this study. These are important points which we now more clearly address or acknowledge.

- 1. Differences between demographic groups in the likelihood of water contact resulting in infection risk due to diurnal cercarial activity:** In Table S6, we report the percentage of water contacts, separated by activity, that occurs at the time of peak cercarial shedding, defined here rather conservatively as 10 am to 3 pm, to investigate if the time of day of water contact differs between demographic groups, as you suggest. We find that among females, 33.7% of all water contact due to washing clothes with soap, 19.8% of all contacts due to collecting drinking water, and 42.6% of all contacts due to washing jerry cans occurred at the high-risk time of day. Among males, 10.8% of all fishing activities were conducted at high-risk times of the day. At the same time, Table S6 shows that there are hardly any gender differences in time of day by activity. In other words, differences between genders in terms of time of day are largely related to the fact that males and females conduct different water contact activities, not that they do the same types of activities at different times of the day. However, male adults had a higher prevalence and intensity of infection than females, despite having a lower proportion of contacts at the peak time of cercarial shedding. Thus, we did not find strong evidence to suggest that differences in time of day were a key factor for explaining gender differences in infection. When we correlated only water contact during peak shedding hours with infection, as you suggested, we found that having water contact during peak shedding hours was not significantly correlated with infection ($\rho = -0.006$, $p = 0.74$). The infection model in Fig S16 allowed for selection from a more extensive set of granular water contact variables, including variables indicating whether an activity was carried out at a high-risk time of day (see Methods, Individual, household, and village human-environmental variables section). Yet, as shown by the AUC performance in Fig 9, inclusion of more granular variables (including time of day) did not boost our ability to predict infection.
- 2. Unaccounted regional differences in water contact patterns:** We have revised our estimation strategy by including district-level fixed effects that can account for unobserved regional differences in environmental risk or any other factors which may differ between our study districts. All results remain robust.
- 3. Potential that our study did not capture the peak transmission season:** As noted in the methods section, we conducted the study in the dry season because we knew from our local counterparts that this is when water contact is highest as people cannot rely on collected rainwater and seasonal ponds have dried up. Thus, we believe that we captured the peak season of water contact. This is also supported by a previous meta-analysis we conducted (Reitzug et al. 2023 in PLoS NTDs), which included 101 studies reporting on associations between schistosome infection and water contact and found that dry-season water contact was most strongly associated with infection. Cercarial abundance may also vary between seasons. The Andrus et al. paper you mentioned found a slightly higher infection prevalence in snails on Lakes Albert and

Victoria in the wet season compared to the dry season. But this difference was relatively small (4-5 percentage points) and not statistically significant. Forthcoming research from SchistoTrack—which is beyond the scope of this current manuscript—suggests that in our study areas, transmission is high throughout the year. Both *B. sudanica* and *B. stanleyi* on Lake Albert, and *B. sudanica* and *B. choanomphala* on Lake Victoria, are very efficient vectors. The only difference is that *B. sudanica* is the main vector during the rainy period on both lakes, while *B. stanleyi* and, to some extent, *B. choanomphala* are major vectors in the dry season, suggesting that there may be no large seasonal changes in transmission. However, detailed longitudinal water contact and malacological data would be required to comprehensively assess when transmission is highest.

4. **Underreporting of children’s water contact:** You are right to point out that underreporting remains one possible shortcoming of the survey-based collection of water contact data. However, as shown in Fig S11, the age distribution in the ‘reference standard’ direct observation data is similar to self-reported water contact data, suggesting children were not significantly less represented in the survey data compared to direct observation data. Thus, while it remains possible that specific activities are not always accurately reported, for instance, if a child fetches drinking water and also goes for a swim, we believe that given the overall agreement between self-reported and observation data for our main outcome of having water contact would still be correct for this individual even if not all activities are captured.
5. **Additional unaccounted factors:** We concur that there could be additional unaccounted drivers at play. However, our models, especially considering we have a behavioural outcome, fit the data very well (AUC = 0.78, which is higher than the AUC for infection, which was AUC = 0.69), indicating that even in the presence of unaccounted individual-level or household-level factors, we have a substantial ability to predict water contact and in fact better ability to predict water contact than infection. As you point out, there is considerable complexity in water contact dynamics, which we seek to understand here. We emphasise the need for future work that draws upon the factors we identified and weighs contact by relevant predictors of environmental risk to generate validated exposure indices. It remains possible that environmental drivers, such as site-level vegetation, play a role in our setting. However, malacologists in our study also searched both fixed and floating vegetation for snail intermediate hosts. Crucially, while Wood et al. (2019) and Jones et al. (2021) have very detailed environmental information, they lack detailed information about site usage of individuals, which means they rely on the assumption that people are using sites within their village. In our analysis, we use the type of landform at the nearest water site as an indicator of how ecological characteristics affect water contact behaviour, adding to our understanding of the interplay between environmental factors and human behaviour. For one of our additional analyses in Fig 9 and Fig S15, we incorporated detailed environmental variables, including snail abundance and infectivity at the nearest site to the household, into the variable selection process. However, we found that this model had no improved performance over our main infection model, suggesting that including environmental risk factors did not have a major impact on our ability to predict infection.

Reviewer 1, comment 5: I report below a series of more or less minor comments. line 57-58: consider to break the long sentence by putting a period after households (line 57), and starting the new sentence with “In addition, there are concerns...”

Response: We have added a period, as you suggested.

Reviewer 1, comment 6: line 93: “water contact data were collected.” This is where I think that, if possible and according to journal guidelines, it would increase clarity if you moved here the first paragraph of the “Exposure” section.

Response: Thank you. We added more details about how we defined and collected information about water contact in the last paragraph of the introductions section for more clarity.

Reviewer 1, comment 7: Line 107: <current water contacts>. Why current? The water contacts of the week? As I mentioned above, current water contacts in the dry season might when snails prevalence is low might less risky than when water contacts occurs in the wet season (which might help to explain the lack of correlation between water contacts and infection)

Response: Based on your comment above, we clarified our definition of water contact in the main text. We refer to ‘current water contact’ because, as shown in this paper, water contact may vary substantially over the life course, so we do not mean to imply that someone’s current typical water contact necessarily reflects their past water contact or their exposure history.

Reviewer 1, comment 8: Line 110: <high intensity water contact>, not sure what the authors mean with high intensity? Long duration? More than 20h/week? Or entire body vs only feet? Who are these “few individuals” fishermen, boys, or distributed more or less randomly among the various demographics?

Response: By high intensity, we mean either a high duration or high frequency of water contact. We reworded this formulation to make this clear and dropped mention of the word ‘intensity’ which was ambiguous.

Reviewer 1, comment 9: Line 129: I am certainly a bit picky here, but how do you have a water contact that does not occur around or close to a water body? Where do the remaining 20% of the water contact occur?

Response: Thank you for this comment. We have reworded this statement to make it clearer. What we intend to convey is that in this study, among all individuals with water contact (which is our denominator here), 80% lived within 0.34 km of a water site, and only 20% of all individuals who reported having water contact lived further than 0.34 km from a water site. Fig. S2 provides further detail on this by showing the distribution of water contacts and households relative to the distance to the closest water site.

Reviewer 1, comment 10: line 134: I suggest starting the sentence “We used generalized additive models” in a new line. Also the authors might consider to add “mixed-effect” to generalized additive model (GAMMs).

Response: As explained further in detail in comment 26, we rely on fixed effects in the main regression models to account for district-level differences in ecological and other factors. Thus, to be consistent with this fixed-effects strategy, we do not use mixed effects here.

Reviewer 1, comment 11: Line 135: maybe I am wrong here, but somehow, I tend do associate “prevalence” to infection, so it sounds weird to me to talk about “prevalence of water contact” although semantically correct (unless the author specify: <... (i.e., fraction of the studied population that has one or more water contacts per week)>, or something like that). This is true also for line 152

Response: We agree with you that the term ‘prevalence of water contact’ may seem somewhat odd. We have replaced it with phrases such as ‘proportion of individuals with water contact.’

Reviewer 1, comment 12: Line 147: <The duration of water contact...>: I think that here the authors meant the frequency, not the duration.

Response: You are correct. Thank you for catching this mistake.

Reviewer 1, comment 13: Line 160-162: if possible, add a reference to support the statement on the peak time for cercarial shedding rate.

Response: Following your suggestion, we added a reference to support this claim.

Reviewer 1, comment 14: Line 174 and following. I got a bit confused here, as I think (but I might be wrong) that BVS is used in association with the GAMMs, but it is not explicitly said so in this paragraph, so it seems that BVS is used to thin the number of variables before running any model, which I don’t think it is the case (unless I am plain wrong).

Response: Thank you for this comment. We have sought to clarify this throughout the results section by adding more detail and explicit headings to ensure that our methods are clearly explained. We first use GAMs to describe the nonlinear relationship between water contact and infection over age and to inform the modelling approach in multivariable regression models. To select variables for the regression models, we use an extended set of covariates (including age). However, because splines would not provide interpretable estimates of age effects on water contact and infection, we opted for using a polynomial term (age²) to allow for nonlinear relationships of age with water contact and infection, respectively.

Reviewer 1, comment 15: Line 190 and following: is the main exposure, logistic model where the dependent variable is whether an individual had one or more water contacts?

Response: In response to your previous comment, we have clarified in the last paragraph of the introduction as well as in the methods section that the outcome here is having one or more water contacts per week.

Reviewer 1, comment 16: Line 198: what type of contamination? Cercarial contamination, chemical contamination (soap foam, plastic pollution), or simply turbidity/murky waters?

Response: Thank you for highlighting this issue. We have revised this phrasing to clarify that we refer to contamination as being observed human stool at the water site.

Reviewer 1, comment 17: Lines 247-248. As this paragraph is about infection, I would turn this sentence reporting first the information on number of variable for infection status (7), in contrast to the 10 variable used as predictor of water contact.

Response: Thank you, we have followed your suggestion and switched the order of reporting.

Reviewer 1, comment 18: Line 272: this sentence on endogeneity is a bit obscure to me, can the authors help the reader (or just me) to understand what this is about.

Response: Thank you for raising this issue. We apologise for the confusion. This line was no longer relevant to the revised manuscript and has been removed.

Reviewer 1, comment 19: Line 333: as I mentioned, here acquired immunity comes out of the blue, with no context why it is important (i.e., because it takes time to mount) and what are the implications (see comments above),

Response: Thank you! We now provide more context on the implications of our findings for the role of acquired immunity in the third discussion paragraph.

Reviewer 1, comment 20: Lines 336-337: (and what else?), the authors might consider to use: - or something like that.

Response: We have substantially revised the paragraph about gender. The sentence you refer to has been removed completely.

Reviewer 1, comment 21: Line 340: after <... than males> consider to add: in agreement with [REF] >

Response: This sentence has also been removed completely.

Lines 347-349: I do not understand what the authors mean with: <... by independent factors that drive gender difference in exposure, but not gender itself>.

Response: We have tried to clarify that the complex effect of gender on water contact is mediated via higher involvement of women in domestic activities and lower involvement of women in occupational activities. This is detailed in the fourth discussion paragraph. The sentence you are referring to has been removed.

Line 345: It seems to me incorrect to talk about <key risk factors for water contact>, I consider them <driving factors>, not risks, as water contact is not a risk per se, it is essential for the life of these populations, although I agree that any water contact in parasite contaminated waters accrues the risk to acquire further infections.

Response: Thank you for pointing out this issue. We have replaced 'key risk factors' with key predictors.

Line 357: what does it mean "having primary-school education"? "to have a primary school degree" or "to attend school"?

Response: We renamed the education variable to clearly indicate that this variable refers to the highest level of education attained/completed.

Reviewer 1, comment 22: Line 360-361: personally, I consider very positive that exposure reduction intervention have been targeted at school age children as these are the age classed with the highest prevalence of infection, so, the most vulnerable one, regardless of their actual contact rate. I think that the goal of MDA is to reduce morbidity, not necessarily to target the most exposed classes.

Response: We have revised the discussion section and point out that our study supports the need for continued MDA in school-age children, as children have high infection prevalence despite low levels of water contact. At the same time, the high levels of water contact in adults support broader targeting of exposure reduction interventions at all age groups.

Reviewer 1, comment 23: Line 361-362: as this study does not solve the conundrum of a lack of correlation between water contact and infection (so, we do not understand how people get infected), I think that this is a bit of an overstatement and my personal suggest is to remove or down tone this sentence.

Response: Thank you. We agree with you. This sentence has been removed.

Reviewer 1, comment 24: Line 406: a 3.4% reduction every 100 meters means a nearly ~30% reduction the fraction of the study population having contact with water in 1km, and a ~65% reduction in 3 km, which is remarkable and worth to emphasize.

Response: You are absolutely correct that our findings imply a 34%-point reduction per 1-km increase and we have added this statistic as you suggested to highlight the magnitude of the distance decay in paragraph six of the discussion section.

Reviewer 1, comment 25: Line 408: is 1.9% the reduction in infection prevalence every 100 meters (within 1km from the shoreline – just half of the reduction rate in pr) or in 1 km? in the former case (as I think), I think that the sentence is a bit misleading. In the latter case, I agree that, although significant, it would be truly a negligible reduction, which means that everything in the first km has the same chance of being infected.

Response: We now clarify that we found a 1.9-percentage-point reduction for every 100-metre increase in household distance between 0 and 1 km (approximately half the reduction we observed in water contact), which translates to a 19-percentage-point reduction. We have added this information to highlight the magnitude of the distance decay in paragraph six of the discussion section.

Reviewer 1, comment 26: Lines 593 on: kudos to the authors for the sophisticated statistical analyses (and please, clarify whether the three regions are used as fixed effects and if villages (random factors) are clustered within the region and the study population clustered within villages?)

Response: Thank you for this comment. We have added more details about our statistical methods in the methods section. We now use district-level fixed effects and show the coefficients for those fixed effects in all our main and supplementary models, enabling readers to see if there were any differences between districts in terms of water contact and infection, after accounting for all our covariates.

We decided to use fixed effects, rather than random effects, for two reasons. First, one assumption required for random effects models is that unobserved factors affecting the dependent variable are not correlated with the observed explanatory variables. This can be formally tested with a Durbin–Wu–Hausman test, which compares the consistency of an estimator using random effects against an alternative estimator using fixed effects. The test statistic from the Durbin–Wu–Hausman test for the water contact model (village-level random effects + district-fixed effects versus district-fixed effects only) provided strong evidence ($\chi^2 = 137.46$, p-value < 0.01) that random effects models would have provided inconsistent estimates. Second, one of the main aims of the paper was to compare coefficients across models, which is why we wanted to use either random or fixed effects consistently throughout. Random effects partition the total variance into variation within groups versus variation between groups. Because the part of the variation absorbed into the random component of the model would differ between our main outcomes—water contact and infection—the use of random effects would make it more challenging to compare coefficients across models. Therefore, we opted to use fixed effects in all our models, thereby controlling for all unobserved factors, such as different types of waterbodies, tribal structures, or anything else that may vary across districts.

We also note that when we conducted a Durbin–Wu–Hausman test for the infection model, we found that a random effects model would have provided consistent estimates ($\chi^2 = 3.6994$, p-value = 0.99). However, when we analysed model residuals from the fixed-effects model, we found that the intra-class clustering coefficient (ICC) was just 0.017, indicating very little unaccounted household-level variation. This means that our household-level factors were likely sufficient to account for household-level variation in the infection models.

We also want to stress that we accounted for household clustering by using clustered standard errors at the household level. This was important due to our sampling design, where we sampled one child and one adult per household. Additionally, due to our reliance on household location for analysis, and critically because the household head reported their own water contact as well as the water contact for children, the possibility of correlated responses within the household had to be accounted for using clustered standard errors.

Reviewer 1, comment 27: A last consideration: I do understand that the confidentiality for the data prevents to make them public. I still think that the scientific community (and especially graduate students in training) would learn a lot from the R scripts the authors used (maybe we a randomized subset of data, where the three studies locations and the village names are anonymized) – I would encourage the authors to make their R script public, either as supplementary information or on GitHub.

Response: We include all our scripts in the revised version of the paper and will make all our code and demo data to enable the smooth running of the code publicly available on Figshare upon publication, enabling users to rerun our entire modelling pipeline.

Reviewer 1, comment 28: I trust the authors will be able to address these points, and I apologize in advance for the cases in which I have misinterpreted authors' intentions and what was clearly reported in the manuscript.

Response: Thank you again for your detailed and thoughtful comments that have helped us to improve the manuscript. We addressed all comments and made revisions or clarifications throughout the text.

REVIEWER #2

Reviewer 2, comment 1: This is a nicely written paper, summarising a very comprehensive and carefully executed study. The authors are to be commended. I only have a few comments:

Response: Thank you for the positive review, and suggestions for improving the manuscript. We have incorporated your feedback.

Reviewer 2, comment 2: 1. The novel implications/recommendations for improved schistosomiasis control strategies arising from this research need to be more explicitly explained. I would suggest adding this as a distinct section near the end of the paper, and summarised in a sentence or two in the abstract. In doing so, please explain to what degree these recommendations may be specific to the case study sites in Uganda versus universal.

Response: Thank you for this comment. In the last two discussion paragraphs, we now more clearly outline the implications for control strategies. As per your suggestion, we also added a sentence highlighting that while we believe that our findings have high internal validity, future research is needed to investigate whether our findings apply to settings other than the one we studied.

Reviewer 2, comment 3: 2. The assumption seems to have been made that all of the studied water bodies are of equal risk in terms of cercarial contamination. Is there any evidence to support this?

Response: Thank you for this comment. We agree that water body type and geographical variation are important to consider. This study accounts for possible variation between districts with different types of waterbodies by using district-level fixed effects, which account for unobserved variation in cercarial contamination or other factors that may differ between districts. Although not the focus of this paper, as we aim to characterise water contact behaviour, we also conducted malacological surveys as part of the SchistoTrack study in 2022, concurrently with the baseline (sampling was conducted at 40 sites in Pakwach, 44 sites in Mayuge, and 59 sites in Buliisa). Districts had different dominant landforms: Pakwach had lakes/marshes, Buliisa had lakes/beaches, and Mayuge had lakes, beaches, and marshes. The two dominant snail species were *Biomphalaria sudanica* and *B. stanleyi*. In terms of the number of snails per site, we found on average eight *B. sudanica* snails per site (IQR 0-21.2) in Pakwach, 115 *B. sudanica* snails per site in Mayuge (IQR 1.5-188), and zero *B. sudanica* snails per site (IQR 0-4.5) in Buliisa. We found zero *B. stanleyi* snails per site (IQR 0-0) in Pakwach, none in Mayuge (as the species is not present on Lake Victoria), and zero *B. stanleyi* snails per site (IQR 0-1) in Buliisa. The median percentage of shedding snails and the interquartile range for shedding snails were zero across all districts.

Reviewer 2, comment 4: 3. In terms of fishing, does this tend to occur near the shoreline, or out in the lakes? Snail and cercariae levels would be expected to be higher near the shoreline.

Response: We agree that all fishing types are not of equal risk. We have added a sentence in the third-to-last discussion paragraph highlighting that we focused on occupational fishing, which is likely different in terms of exposure risk compared to recreational or occasional fishing. Offshore fishing is mainly associated with wealthier fishermen using larger boats, primarily targeting Nile perch. In contrast, women, children, and poorer fishermen with smaller boats who primarily target silverfish are limited to shoreline fishing. However, upon landing, even deep-water fishermen wade and bathe in shallow water, so there may not necessarily be a stark difference in the risk of infection between different types of fishing. We also found a high correlation between fishing occupation and self-reported fishing activity

(rho=0.68, p<0.01). Fishing practices also vary between different geographies, which we account for using fixed effects in our regression models. Particularly in Lake Albert, even far from the shoreline, there are many shallow areas where people bathe, fish, and rest. Despite the heterogeneity in fishing practices, this variation was not so pronounced that it obscured an association between fishing and high infection intensities, as we were still able to find a positive relationship between fishing and water contact and infection (Figs 7-8).

Reviewer 2, comment 5: 4. In terms of washing, do all people use soap when doing so? If not, it might be interesting to compare infection levels of those using soap and not.

Response: Thank you; this is an important question to address. We collected data on both washing clothes with and without soap. In Table S3, we report on these activities. Among our participants, 16.3% reported washing clothes with soap, and 5.3% reported washing clothes without soap, suggesting that roughly three-quarters of all individuals who wash clothes use soap. However, these categories were not mutually exclusive. As we now report in the main text, individuals who washed clothes with soap were as likely as other participants to report washing clothes without soap (5.8%, 28/481, versus 5.1%, 124/2386, $p = 0.67$). We also compared infection levels between those who washed clothes with soap and those who washed clothes without soap (restricted only to people who had water contact). We found no significant difference in infection status based on soap usage, according to a Chi-squared test of independence ($\chi^2 = 0.32$, $df=1$, $p=0.57$, $n = 1339$). Due to clear gender roles in water contact activities, males less frequently reported washing clothes with soap (males 8.7% and females 23.4%, $p<0.01$). Males also less frequently reported washing clothes without soap (males 2.9% and females 7.3%, $p<0.01$, Table S3). Although there were clear gender patterns, the statistics also show that soap use is not entirely gender-dependent. The complexities of soap usage patterns make it difficult to clearly attribute different risk profiles to different activities. The uncertainty around how protective soap is against schistosome infection is also highlighted by a recent experimental laboratory study (Zhang et al. 2024 in PLoS NTDs), which found that only very high soap concentrations (the highest concentration of 1000 mg/L tested in the study) resulted in 100% cercarial death at 5 minutes. Thus, establishing the efficacy of soap under real-world conditions remains challenging where soap concentrations will be quite variable, and the use of soap will vary dependent on behaviour, cultural norms, and finances to buy the soap. We also note that we have included both washing with and without soap as candidate variables in our extended model. As described in the legend of Figure S9 and Fig. S16, the number of hours spent washing clothes with soap was selected as an additional variable but was not significantly associated with infection, while washing clothes without soap was somewhat counterintuitively associated with lower odds of infection (OR 0.66; 95% CI 0.46 – 0.94).

Reviewer 2, comment 6: 5. The way that the two sentences on lines 147-150 are worded makes them seem directly contradictory - 'The duration of water contact was higher for females than males...Females had a lower duration of water contact than males...' Suggest rewording. Check carefully for other contradictions elsewhere in the paper.

Response: Thank you for spotting this mistake. The first mention was meant to be 'frequency' instead of duration. We have corrected these lines to resolve the contradiction.

Reviewer 2, comment 7: 6. Related to my first comment, it would be a shame if a policymaker read this paper and came away with the conclusion that addressing water contact is not important, since the determinants of exposure are not the same as the determinants of infection. Ultimately all infections were caused by exposure at some point in the lifetime of the infected person via some kind of water contact activities, so this should be emphasised.

Response: Thank you for pointing out this important issue. To address your comment, we added a sentence at the very end of the first discussion paragraph emphasising that our findings should not be taken to imply that water contact is unimportant for infection as all infections are ultimately caused by cercarial exposure via water contact.

REVIEWER #3

Reviewer 3, comment 1: The authors investigated potential correlation between self-reported frequency and timing of water contact activities (dependent variable) and a host of demographic, socio-economic, WASH, and geographic variables (independent variables) among ~1400 households across 38 rural fishing villages in Uganda. The same independent variables were used to assess a potential correlation with infection as measured by Kato-Katz (or an antigen test). Presence of snails and infected snails were considered as additional predictors. Self-reported data on water contact were further compared to directly observed behaviour in 12 villages. The study is a very valuable effort to better understand the link between WASH variables, water exposure and (age patterns) in infection. Overall, I'm positive about the manuscript, but think it would benefit from some additional clarifications, reformatting of certain figures, rephrasing of key messages, and additional analysis of infection intensity data.

Response: Thank you for providing such a helpful and detailed review. We have incorporated your suggestions throughout, which have greatly strengthened the manuscript.

Reviewer 3, comment 2: Regarding: "For estimating the force-of-infection, current mathematical models assume age-specific prevalence to be proportional to age-specific trends in current water contact [22] and do not always account for individuals who remain infected for years without current exposure due to the long lifespan of the schistosome parasite [23]."

The first part of the statement about reliance of models on age patterns in water contact may be true for some models, but not all. In some models (e.g., Imperial College London), the age patterns in exposure are estimated from the observed patterns in intensity and prevalence of infection by age. Regarding the second part of the statement about not always accounting for the long lifespan of schistosomes: I'm not aware of any mathematical models that do not account for this. The lifespans of the adult worm pairs in the human host are always explicitly simulated as this aspect dictates the population dynamics of SCH for a large part.

Of note is that all existing mathematical models SCH assume that acquired immunity, if any, does not affect the probability of parasite establishment on exposure. As such, all model-estimated and predicted age patterns in exposure and infections levels are conditioned on this assumption.

Response: We agree, and our writing may not have as clearly conveyed the same point you are trying to make as clearly as you have made it. We have used phrasing inspired by your comments to reword our statement in the third paragraph of the introduction section which now reads: "For estimating the force-of-infection, current mathematical models infer age-specific water contact levels from infection trends over age^{22,23}. As such, the models simply scale levels of water contact to be proportional to age-specific infection levels."

Reviewer 2, comment 3: Regarding: "This study was designed to detect a minimum effect size of 8% with an unevenly exposed population and a household design effect of 1.136 at 97.5% power."

Please clarify what the expected minimum effect size is referring to. A difference in water exposure and/or infection prevalence/intensity? Between which groups? And what was the (seemingly small) household design effect of 1.136 based on? Was this design effect intended to capture correlation of observation within households and/or within villages?

Response: This statement has been removed in favour of a reference to Anjorin et al. (2023), who provide details on the sample size calculation for the SchistoTrack study. Briefly, the statement referred to an 8% difference in infection prevalence, and the small household

clustering design effect was calculated from previous studies conducted in our study areas (Lamberti et al., 2021, PloS ONE). A small correlation within households is to be expected, given that we sampled only one adult and one child, and the burden of infection differs substantially between these groups.

Reviewer 2, comment 4: Please clarify whether single or duplicate Kato-Katz was done (or something more elaborate, e.g. multiple days).

Response: Thank you. We added details in the *S. mansoni* infection section within the methods: “We prepared duplicate, thick smear slides from a single, stool sample produced on the morning of the slide preparations. Senior technicians re-read a random sample of 10% of all slides for quality control.”

Reviewer 2, comment 5: Was there a particular need to round age to the nearest year? This is emphasised in both the methods section and the figures. Was there a particular reason to explore age polynomials up to the 4th order? E.g., did you expect two peaks in infection or exposure levels across the range of sampled ages? Also, this information would be expected in the statistical analysis section and not the data collection section. Moreover, later on in the “statistical analysis”, the use of generalised additive models (GAMs) is mentioned, with explicit mention of the use of knots to model non-linear age patterns. This seems to be a different approach than the use of simple polynomial terms. Please clarify.

Response: In setting up the cohort, the definition of age was explored. The exact birth date is not always recorded or known by participants, so age was only recorded to the nearest year. We appreciate that our approach to specifying age requires clarification and apologise for any confusion introduced. GAMs were only used in exploratory analyses to understand the functional form of age. We now limit the polynomials to age^2 only, as we agree that going beyond this was not supported by the shape of the GAMs. In the results, we descriptively describe the age^2 term to indicate that there was a significant change in the slope at older ages. We decided to use polynomials in the main models because the shape of water contact and infection over age in GAMs suggests that age and age^2 are sufficient to approximate the relationship, and polynomials require fewer degrees of freedom than if we had used splines in the regression models.

Reviewer 3, comment 6: The authors used Pearson correlation coefficients to assess correlations between water contact and infection status across participants. First off, this implies that these variables are assumed to be correlated in a linear fashion, which is arguably doubtful (e.g., why not use Spearman correlation).

Response: Spearman correlation and Pearson correlation are identical when all data are binary, as was our case, but we have changed throughout to Spearman’s rho.

Reviewer 3, comment 7: When describing the water contact model, the authors state that “For all regression models, we clustered standard errors at the household level to account for our sampling design where households were randomly selected, but not individuals.” How was this handled in the models for infection? There, I would expect additional clustering of residuals at the village level, as infections in the same villages are connected via their shared environmental reservoir of infection.

Response: Thank you for asking for this clarification. We used the same approach for infection and have added additional details in the statistical analysis section within the methods to ensure we provide enough information to make this clear. We investigated clustering of residuals for the infection models by calculating the intra-cluster correlation coefficient (ICC) of the model residuals at the village level. The ICC of the residuals for our

infection status variable (measured by an ‘empty model’ with just village-level random effects) was minimal, at only 0.017. This indicates that while we have not controlled for all possible village-level factors, the remaining unexplained village-level variation was very low.

Reviewer 3, comment 8: Data availability: the authors state that the “Data is not publicly available due to data protection and ethics restrictions related to the ongoing nature of the SchistoTrack Cohort and easily identifiable nature of the participants. Code can be shared upon reasonable request to the study authors.” The offer to share only code on reasonable request is not in the spirit of open science. Code can be stored separately and can be easily shared via open-source repositories like GitHub or GitLab. Also, I don’t see why the data cannot be made publicly available. Why can’t the data be pseudo-anonymised, removing personal identifying data and geolocation data and replacing individual, household and village identifiers with codes to which the authors themselves only have the codebook to identify individuals? I think making both the code and data available is important, in particular given the data richness and the many ways in which the research question could have been answered with different statistical methods and approaches.

Response: Thank you for this comment. Following your suggestion, we now include all our scripts in the revised version of the paper and will make all code as well as dummy data publicly available on Figshare upon publication of the paper, enabling users to rerun our entire modelling pipeline.

Reviewer 3, comment 9: Why not also model intensity of infection and only presence of infection? Intensity data contains much more information than prevalence data.

Response: We modelled intensity of infection, though only as a binary indicator in Fig. 8 to show the WHO category of heavy infection intensity. We now provide an infection intensity model using a negative binomial regression model to predict infection intensity in Fig. S10. We used likelihood ratio tests to compare Poisson models with negative binomial models and tested whether count models were appropriately modelling zeros in the outcome using the ‘check_zero_inflation’ function in the R package ‘performance’. We found that a negative binomial model fit the data better than a Poisson model and was able to adequately model zeros.

Reviewer 3, comment 10: Up to line 171, the authors describe a lot of summary statistics for many different variables. At this point during the reading, I was not clear on whether these variables had already been selected or not with the Bayesian variables selection, or is this a description of everything that was measured? Only after line 171, the reader gets to Figure 6, which provides an overview of the variables that made it through the selection procedure. It would be helpful if the reader can be guided a bit more through that.

Response: Thank you for pointing out where sentences could better guide the reader. We have adjusted the section heading to clarify that the first part, up to the ‘Relevant human-environmental dimensions of water contact’ section, is descriptive and precedes the variable selection. We focused on describing water contact here (its frequency and duration), gender and age patterns, and distance, as these were key variables discussed previously in the introduction and other published literature as important predictors of water contact. We also wanted to provide the metadata in case it proves useful in the future for mathematical models or for meta-analysis.

Reviewer 3, comment 11: For all figures with probabilities or percentages on an axis: please use increments of 20% instead of 25% for tick marks. Interpreting data points or estimates

half-way a 25% tick mark is very awkward. And please consider adding grid lines at 10% increments. These adjustments would significantly improve the readability of figures.

Response: We have incorporated your comments to improve the readability of the figures.

Reviewer 3, comment 12: “Our study population was characterised by a high prevalence of self-reported water contact with open freshwater bodies (46.7%, ...)”. Please clarify what is meant here by water contact. Does this mean that 53.5% of the participants said they never has any water contact? Or perhaps water contact at some frequency less than XX? Something else? Please help the reader understand what you quantified here. The next sentence refers to “current water contact”, but it is also not clear what is meant by that without going to the questionnaire. Although the authors state that “Detailed variable definitions and characteristics of the 2867 participants are presented in Tables S1-2”, I would expect the definition of “water exposure” as one of the main outcomes to be explained and described in the main text and not in a supplement.

Response: Thank you. We have added the definition of how we define water contact in the last paragraph of the introduction to make this clear to the reader without having to refer to the methods. Your interpretation is indeed correct that 53.5% of the participants reported that they do not typically have any open water contact during a one week period.

Reviewer 3, comment 13: There seems to be a numerical discrepancy between the two following statements made by the authors:

- Line 105-106: “Our study population was characterised by a high prevalence of self-reported water contact with open freshwater bodies (46.7%, ...)”
- Line 129-130: “Water contact was highly concentrated around waterbodies as 80% of all participants with water contact (and 64.3% of all participants) lived in households within 0.34 km Euclidean distance of a water site”

Given the first statement, the 80% in the second statement would translate to $80\% * 46.7\% = \sim 37\%$ “of all participants”, whereas it says “64.3% of all participants”.

“3.4% absolute reduction” consider “reduction of 3.4%-points” or “...3.4 percentage points”.

Response: Thank you for highlighting how our wording could be confusing. We intended to convey that 80% of all participants *with water contact* lived within 0.34 km Euclidean distance of a water site, and 64.3% of all participants resided within 0.34 km Euclidean distance of a water site. However, these two statistics are different and cannot be directly multiplied together. Your calculation indicates that 37.3% of all people in our study have water contact and live within 0.34 km Euclidean distance of a water site, which is correct (you calculated cell 1,1 of the two-by-two matrix). Hopefully, the table below clarifies how these statistics relate to each other. They were not contradictory but were confusing as previously written and we have rewritten them.

We have also revised the statement regarding the 3.4% absolute reduction and now refer to this as a reduction of 3.4 percentage points. Any similar types of statements have been revised throughout the manuscript.

Two-by-two table of proportions of the population with/without water contact by waterbody distance

	Water contact	No water contact	Row totals
Within 0.34 km of a water site	0.3736 (0.467*0.8)	0.2694	0.643
>0.34 km of a water site	0.0934	0.2636	0.357
Columns totals	0.467	0.533	

Reviewer 3, comment 14: Figure 7: because the estimated odds ratios are plotted on a linear scale, most of the graphics are almost impossible to interpret. Many are very close to 1.0 or have extremely narrow confidence bounds. Sometimes this is also due to the chosen scale at which predictors are expressed (e.g., age, which could also have been expressed as “decades”, which is just as arbitrary as “years”, but might yield more legible estimates). A first step towards addressing this would be to plot the estimates on a logarithmic scale with equally spaced ticks at OR = 0.5, 1, 2, 4, etc.

Response: We have followed your recommendation to log-transform the axis of all forest plots.

Reviewer 3, comment 15: The interpretation of the coefficient for age² seems off: “The age² term (odds ratio (OR) 0.9978; 95% CI 0.9975 – 0.9982), indicated significant nonlinearity but only a 0.0022% decrease with each one-year age increase.” This coefficient is the change in log-odds per unit change in age², so not a “one-year increase”. As mentioned above, the scale of this coefficient is this small because of the choice of scale for age (and consequently the enormous scale for age²). More generally, it’s not very meaningful to try to interpret the coefficients for the linear and square age terms separately; it’s the set of the two together that explains the pattern. This reminds me, no mention is made of the “knots” (which are suggestive of the use of splines) that are mentioned in the methods section as a way to model the non-linear age patterns. The additional presence of such a component in the regression model would make the interpretation of the individual coefficients for age and age² completely meaningless. If splines are indeed used, the age patterns are best described in a qualitative manner, with reference to Figure 4.

Response: Thank you for this comment. As mentioned above in our response to comment 5, we modelled age by only including age and age² terms precisely for the reason you mentioned – that if we had included splines, we would not be able to interpret coefficients and compare across models. We agree and now interpret the age and age² terms together and lean towards a qualitative interpretation of age².

Reviewer 3, comment 16: Why was village-level SCH prevalence dichotomised as a predictor? For age, you explicitly argue to use at a continuous predictor so as to not lose information.

Response: We discretised village-level prevalence of schistosomiasis into three levels that correspond to the endemicity settings defined in the 2022 WHO guidelines on the control and elimination of schistosomiasis: 0–10% prevalence by Kato-Katz corresponding to a low endemicity setting, 10–49% prevalence corresponding to moderate endemicity settings, and 50% prevalence corresponding to high endemicity. However, we ended up with a binary variable because there were no low-prevalence villages in our study. As these categories inform WHO treatment strategies, we thought it was important to investigate whether water contact also differs between these settings, as it has been suggested that high endemicity settings may be characterised by higher levels of water contact. Thus, even though we agree with you that this characterisation may be somewhat crude, we believe it was important to analyse the data in a way that would allow us to link our findings to the current WHO guidelines and provide suggestions for future intervention research. We note the rationale for these cut-offs in the *S. mansoni* infection section within the methods.

Reviewer 3, comment 17: Note for the discussion section: several of the independent variables in the regression analyses are measured with considerable uncertainty/sampling error. The larger this (random) measurement error in an independent variable, the more one

underestimates its correlation with the outcome (dependent variable) due to regression towards the mean.

Response: We agree with this comment for the water contact variables though believe it to be less of importance for the most relevant malacology variables such as water type (river, lake, etc.). We now clearly share this as a potential limitation of the study at the end of the seventh discussion paragraph. “Still, there remains a need for future water contact ascertainment studies that match individuals directly with more granular measurements, and when such ascertainment is completed, we suggest rerunning our models with regression dilution to investigate the influence of possible measurement error. For future research, more granular measurements of frequency and duration of water contact should be investigated using wearable GPS loggers..”

Reviewer 3, comment 18: Regarding: “In terms of WASH, we found that in villages where all study households used a safe drinking water source (taps/boreholes), individuals had a 65% lower likelihood of having water contact compared to people living in villages where all study households used surface water (OR 0.35, 95% CI 0.24 – 0.52).”

This seems an unnecessary level of data aggregation for access to a safe drinking water source. Why was this association quantified at the village level and not the household level (e.g., using a 0/1 indicator for use of a safe drinking water source)? Unless there is a good reason (which the authors should then explain), such an unnecessary aggregation poses a risk of unintentionally introducing ecological fallacy into the analysis.

Response: Thank you for raising this point. We carefully considered your comments and agree with your assessment that the aggregation of our household-level safe drinking water variable to the village level could have introduced an ecological fallacy, especially as we found that the household-level variable was not significantly associated with having any water contact. We have therefore removed the village-level safe WASH variable from our model. Instead, we generated household-level variables that capture relevant WASH factors and are much less prone to ecological fallacy: we created a variable indicating whether the household used any public taps or latrines, and variables indicating the Euclidean distance from the household to the nearest tap or borehole and the nearest public latrine. We also carefully revisited all other village-level variables to ensure we did not risk introducing ecological fallacies. Upon reflection, we removed the variables for the number of public taps per village and the number of public latrines per village, as our new household distance variables provide less aggregated and more detailed information on public infrastructure. We also removed the ‘number of roads per village’ variable because it is correlated with lake distance, and we felt that there was no clear mechanism by which the number of roads would affect water contact, which could also introduce an ecological fallacy.

Reviewer 3, comment 19: Regarding: “We found that at-risk groups for water contact and infection differed in their observable characteristics suggesting that transmission/exposure and infection control interventions should not be targeted to a single or the same risk group.” This second part of this conclusion (underlined) is formulated in rather strong words. I had a very strong knee-jerk response of disagreement with this: of course, water exposure leads to acquisition of new worms, but potential regulation of parasite establishment by acquired immunity in humans may well explain the observed difference in age patterns; so why this strong statement about (not) targeting the same or different groups? But I then realised I’m not even sure what the authors mean here. What do the authors mean with “targeted to a single group or the same risk group”? This conclusion is also not substantiated further on in the discussion. Perhaps it would be helpful if the authors formulated their conclusion in terms of what they think “should” happen, rather than “should not”. Also, I think the insight this

study provides into a potential role of acquired immunity in the human host deserves to be mentioned in this first paragraph.

Response: We agree and have removed this statement. The discussion has been revised so that when age-specific patterns are discussed, there is a more thorough exploration of acquired immunity in the third discussion paragraph. We also tone down the language in our concluding paragraph and emphasise that due to the acquired immunity our results further support MDA towards school-aged children.

Reviewer 3, comment 20: “We suggest that the effects of gender on exposure and infection are directly mediated by independent factors that drive gender differences in exposure but not gender itself.” I’m not sure what the authors mean here. Please consider to rephrase. I’m also missing other potential explanations for the absence of an association between gender and infection: (1) infection is defined as positivity/negative whereas intensity of infection (eggs per gram faeces) can still vary between individuals; (2) can the authors really rule out that the (absence of) gender difference is explained by (absence of) gender differences in the type and location of water contact? See also my earlier comment about regression towards the mean with imperfectly measured predictors.

Response: We agree and have removed this sentence. The gender paragraph has been completely overhauled. We now focus solely on the relevance of gender for water contact patterns and refrain from commenting on the lack of association between gender and infection and explanations for this as this is not the main focus of this paper.

Reviewer 3, comment 21: The typesetting of the document made the text rather hard to read: letters of the same words are often squashed together and/or irregularly spaced. For example, see the letter “i” (in general) and the word “infection”.

Response: We apologise for the formatting errors. We have opted to left-align our text in the revised version since we believe that the squashing/irregular spacing may be due to having used the ‘justify text’ option in MS word.